# MECTA: Memory-Economic Continual Test-Time Model Adaptation

**Junyuan Hong**[1]*, **Lingjuan Lyu**[2], **Jiayu Zhou**[1], **Michael Spranger**[2]
[1]Michigan State University, [2]Sony AI
{hongju12,jiayuz}@msu.edu, {lingjuan.lv,michael.spranger}@sony.com

## Abstract

Continual Test-time Adaptation (CTA) is a promising art to secure accuracy gains in continually-changing environments. The state-of-the-art adaptations improve out-of-distribution model accuracy via computation-efficient online test-time gradient descents but meanwhile cost about times of memory versus the inference, even if only a small portion of parameters are updated. Such high memory consumption of CTA substantially impedes wide applications of advanced CTA on memory-constrained devices. In this paper, we provide a novel solution, dubbed MECTA, to drastically improve the memory efficiency of gradient-based CTA. Our profiling shows that the major memory overhead comes from the intermediate cache for back-propagation, which scales by the batch size, channel, and layer number. Therefore, we propose to reduce batch sizes, adopt an adaptive normalization layer to maintain stable and accurate predictions, and stop the back-propagation caching heuristically. On the other hand, we prune the networks to reduce the computation and memory overheads in optimization and recover the parameters afterward to avoid forgetting. The proposed MECTA is efficient and can be seamlessly plugged into state-of-the-art CTA algorithms at negligible overhead on computation and memory. On three datasets, CIFAR10, CIFAR100, and ImageNet, MECTA improves the accuracy by at least 6% with constrained memory and significantly reduces the memory costs of ResNet50 on ImageNet by at least 70% with comparable accuracy. Our codes can be accessed at https://github.com/SonyAI/MECTA.

## 1 Introduction

Many machine-learning applications require deploying well-trained deep neural networks from a large dataset to out-of-distribution (OOD) data and dynamically-changing environments, for example, unseen data variations (Dong et al., 2022; Liu et al., 2023) or corruptions caused by weather changes (Hendrycks & Dietterich, 2019; Koh et al., 2021). Hence, the recent efforts aim to tackle this emerging research challenge via continual test-time adaptation (CTA). The unsupervised, resource-constrained, and dynamic test-time environments in CTA make it a challenging learning problem and call for a self-supervised, efficient and stable solution. Decent examples include Tent (Wang et al., 2021) and EATA (Niu et al., 2022). Early in 2017, Li et al. found that updating batch-normalization (BN) layers with all test-time data without any training greatly improved the model OOD performance. Recently, Tent (Wang et al., 2021) significantly improved the test-time performance by minimizing the prediction entropy in an efficient manner where only a few parameters are updated. More recently, EATA (Niu et al., 2022) improved sample efficiency and evade catastrophic forgetting of the in-distribution data.

While Tent and EATA had achieved impressive gains on OOD accuracy via online optimization, such optimizations are accompanied by large memory consumption and are prohibitive in many real-world CTA applications. Since many devices are only designed for on-device inference rather than training, memory-limited devices, like small sensors, cannot afford CTA algorithms. In Fig. 1, we demonstrate that Tent/EATA adaptation of ResNet50 (He et al., 2016) with a batch size of 64 (the default setting in Tent) costs more than 5 times of memory in `model.backward` as the standard inference on ImageNet-C (Hendrycks & Dietterich, 2019). The large peak memory consumption makes EATA or

---

*Work was done during the internship at Sony AI. Correspondence to Lingjuan Lyu

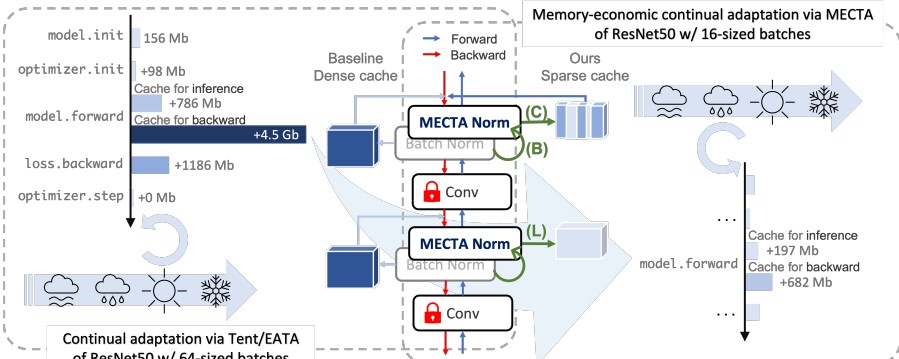

Figure 1: Demonstration of incremental memory footprints brought by each operation on ImageNet and illustration of the proposed MECTA method, which reduces the cache size of gradient-based adaptation. During forwarding, the MECTA Norm (**B**) stabilizes the normalization-statistic estimation via shift-aware moving-average from *small batches*, (**C**) randomly drops caches by *channel* admitting sparse gradient descent without knowing gradient in advance, and (**L**) maintains caches only for *layers* on demand of training.

Tent impossible to be adopted on edge devices, for example, the popular board-computer, Raspberry Pi with 1 Gb RAM, and old-generation of smartphones (Ignatov et al., 2018).

Observing the bottleneck on `model.backward`, a straightforward solution could be reducing batch sizes and model scales (including the number of channels and layers), but there are several obstacles to maintaining model performance simultaneously. First, a large batch size is essential for adaptation (Yang et al., 2022). Second, the amount of information extracted by a deep and wide model is desired for modeling distributionally-robust semantic features (Hendrycks et al., 2020a).

In this paper, we tackle the aforementioned challenges by proposing a novel approach called Memory-Economic Continual Test-time Adaptation (MECTA). As illustrated in Fig. 1, our method is enclosed into a simple normalization layer, MECTA Norm, to reduce the three dimensions of intermediate caches: batch, channel and layer sizes. (**B**) MECTA Norm accumulates distribution knowledge from streaming mini-batches and is stable on small batches and on shifts between domains using a shift-aware forget gate. (**C**) Resembling sparse-gradient descents, we introduce test-time pruning that stochastically removes channels of cached intermediate results without knowing gradient magnitudes. (**L**) The forget gate also guides the layer adaptation: if the layer distribution gap is sufficiently small, then the layer will be excluded from memory-intensive training. Our contributions are as follows.
• *New Problem*: We initiate the study on the memory efficiency of continual test-time adaptation, revealing the substantial obstacle in practice.
• *New Method*: We propose a novel method that improves the memory efficiency of different CTA methods. The simple norm layer structure is ready to be plugged into various networks to replace batch-normalization. The MECTA Norm layer also enables us to stop and restart model adaptation without unused or absent caches for unwanted or on-demand back-propagation. Without forgetting due to removing parameters, our pruning is conducted on cache data for back-propagation rather than forwarding and can greatly reduce memory consumption.
• *Compelling Results*: Our method maintains comparable performance to full back-propagation methods while significantly reducing the dynamic and maximal cache overheads. Given limited memory constraints, our method improves the Tent and EATA by $8.5 - 73\%$ accuracy on CIFAR10-C, CIFAR100-C, and ImageNet-C datasets.

## 2 RELATED WORKS

**Test-time Adaptation** (TTA) aims to improve model accuracy on Out-of-Distribution (OOD) data by adapting models using unlabeled test data. In comparison, traditional learning algorithms train models robustly, e.g., distributionally robust neural networks (Sagawa et al., 2020) or adversarial training (Deng et al., 2023), in order to generalize to OOD data. Early examples for TTA include the test-time training (TTT) (Sun et al., 2020) and its variant (Liu et al., 2021) which jointly train a source model via both supervised and self-supervised objectives, and then adapt the model via self-supervised objectives at test time. Adaptive risk minimization (Zhang et al., 2021), contextual

meta-learning, conditional neural process (Garnelo et al., 2018) train models that can be adapted using inferred context information from test data. Though plausible, these methods are in need of a redesign of the training process. Thus, they are less applicable for many off-the-shelf models pre-trained on datasets that are too big to attain by a small device or are prohibited from unauthorized sharing. To address the problem, training-agnostic adaptation methods are proposed recently, through adapting batch-normalization statistics (Nado et al., 2021; Schneider et al., 2020; Khurana et al., 2022), minimizing unsupervised entropy losses (Wang et al., 2021; Goyal et al., 2022), maximizing prediction consistency over multiple augmentations (Zhang et al., 2022), and classifier adjustment (Iwasawa & Matsuo, 2021). Though these methods are effective, their applicability is limited to a constant test environment in contrast real scenarios.

**Efficient Continual Test-time Adaptation (CTA)** considers the test scenario with dynamically changing rather than static environments and calls for efficient adaptation in place and in time. Wang et al. initiated the continual setting on computer vision by augmented supervision, which however has a large computation hurdle due to multiple times of inference on one sample. The BN-based approaches are efficient alternatives, which only need to update statistics (Nado et al., 2021; Schneider et al., 2020). Yet, it has been shown in recent research that removing covariate shift by BN is not enough for reaching the supervised performance (Wang et al., 2021; Niu et al., 2022). In comparison, Tent is more effective and efficient with one-time back-propagation per sample (Wang et al., 2021). Later, Niu et al. (2022) improved the sample efficiency via selective back-propagation. Despite the advance of CTA in computation efficiency, the memory efficiency is overlooked regardless its importance in on-device adaptation. Plausible directions are on-device sparse (continual) learning (Wang et al., 2022b; Mallya et al., 2018; Mallya & Lazebnik, 2018) or fine-tuning (Jiang et al., 2022; Cai et al., 2020), but how to conduct memory-efficient learning without labels is still unclear. In this paper, we inspect the gradient computation in parameter-efficient adaptation and disclose the main bottleneck for striking the balance between memory consumption of feature caches and learning effectiveness. Based on the analysis, we propose a novel solution to reduce caches by batch, channel and layer reductions without sacrificing performance.

## 3 PROBLEM FORMULATION

Let $P(x)$ denote the data distribution sampled from the set of distributions $\mathcal{P}$, and $P_0$ is the training distribution. We assume that a distributions in $P \sim \mathcal{P}$ is either identical to or is biased from $P_0$ a lot. A neural network model $f_\theta$ parameterized by $\theta$ is pre-trained on the training set $P_0$ by $\theta_0 = \min_{\theta \in \Theta} \mathbb{E}_{x \sim P_0(x)}[\ell(f_\theta(x), c(x))]$, where $c(x)$ is an oracle function for ground-truth labels and $\ell$ is a loss function defined over the sample $x$ and model parameter $\theta$. One example of the loss is the cross-entropy loss: $\ell_{\text{ent}}(f_\theta(x), c(x)) = -\sum_i c(x)_i \log f_\theta(x)_i + \log \sum_i \exp f_\theta(x)_i$ given logits from a $\theta$-governed model $f_\theta(x)$. With the pre-trained model, Tent and EATA continually adapt the model via recursive updating,

$$\theta_t = \text{Optimize}_{\theta \in \Theta_t}(\mathbb{E}_{x \sim P_t(x)}[H(f_\theta(x))], \theta_{t-1}), \ P_t \sim \mathcal{P}, \tag{1}$$

for step $t \in \{1, 2, 3, \dots\}$, where $\text{Optimize}(\cdot, \cdot)$ represents a generic optimization algorithm minimizing the first variable given $\theta$ initialized by $\theta_{t-1}$. Without labels, the entropy function $H(f_\theta(x)) = \ell_{\text{ent}}(f_\theta(x), f_\theta(x))$ resembles the cross-entropy loss with self-supervisions in Tent.

**Parameter-efficient adaptation.** As the efficiency is critical at test-time, we focus on the state-of-the-art efficient solution to the CTA: EATA and Tent, both of which adopt the one-step gradient descent as the optimization strategy in Eq. (1). Thus, $\text{Optimize}_{\theta \in \Theta_t}(\mathbb{E}H, \theta_{t-1}) = \theta_{t-1} - \eta \frac{\partial}{\partial \theta} \mathbb{E}H$ given a learning rate $\eta$. To efficiently train models, we constrain the parameter space in a subspace of the original one, denoted as $\Theta_t = \tilde{\Theta} \subset \Theta$. We follow Tent and EATA to make the parameters trainable in the batch-normalization layers (BN layer).

Now, we give a brief introduction to the batch-normalization layer (Ioffe & Szegedy, 2015). Let the input to the layer $l$ be a *batch* of features denoted as $x^l$ in the real space of dimension $B \times C^l \times H^l \times W^l$, where $B$ is the batch size, $C^l$ is the number of channels, $H^l$ and $W^l$ are the height and width. We use $[N]$ to denote the set $\{1, \cdots, N\}$. For $(n, i, j, k) \in [B] \times [C^l] \times [H^l] \times [W^l]$ and small constant $\epsilon_0$, the BN layer is defined as two sequential channel-wise operations,

$$z^l_{n,i,j,k} = \frac{x^l_{n,i,j,k} - \mu^l_i}{\sqrt{\sigma^2_i + \epsilon_0}} \text{ (normalization)}, \ a^l_{n,i,j,k} = \gamma^l_i z^l_{n,i,j,k} + b^l_i \text{ (affine)}, \tag{2}$$

where $\mu_i^l, \sigma_i^{l\,2}$ are the mean and variance of $x^l$ in channel $i$, respectively. The output tensor $a$ is also called activation. For Tent and EATA, only the affine parameters, $\gamma$ and $b$, are trainable.

## 4 PROPOSED METHOD

In this section, we elaborate on the proposed method that improves the memory efficiency of gradient-based adaptation methods. First, straightforward derivations show that the intermediate representations ought to be stored for computing the gradients of affine layers and therefore forge a huge memory overhead. Suppose the loss on the $n$-th sample is $\ell_n$. Based on Eq. (2), the gradient w.r.t. the $i$-th channel affine weight $\gamma_i$ is

$$\sum_{n=1}^{B} \frac{\partial \ell_n}{\partial \gamma_i^l} = \sum_{n=1}^{B} \sum_{j=1}^{W} \sum_{k=1}^{H} \frac{\partial \ell_n}{\partial a_{i,j,k}^l} z_{n,i,j,k}^l. \tag{3}$$

Therefore, to compute the gradient, each BN layer has to store the normalized representations $z^l$ (*cache*) of the dimension $B \times C^l \times W^l \times H^l$ at forwarding until $\partial \ell_n / \partial a_{i,j,k}^l$ is available. For an $L$-layer network, the inference-only memory consumption of affine layers is $R_{\text{fwd}} = \max_{l \in \{1, \cdots, L\}} B \times C^l \times W^l \times H^l$. In comparison, the corresponding intermediate memory for back-propagation is accumulated by $R_{\text{bwd}} = \sum_{l=1}^{L} B \times C^l \times W^l \times H^l \geq R_{\text{fwd}}$. To reduce the memory overhead, a straightforward idea is to reduce $B$, $C^l$, and $L$ by dropping corresponding entries in $\{z^l\}_{l=1}^{L}$, respectively. Since dropping entries in $\{z^l\}_{l=1}^{L}$ will vanish the corresponding gradients, the strategy will easily break down the learning if not carefully handled. Below, we will discuss the obstacles and our solutions for accuracy-secured cache reduction by $B$, $C^l$, and $L$ dimensions, respectively.

**(Reduce $B$) Adaptive statistic estimation on dynamic distributions.** As a sufficient number of samples are essential for accurate statistic estimation ($\mu$ and $\sigma^2$) per BN layer, reducing samples in a batch will bias the statistics for normalization. In standard or test-time training, exponential moving average (EMA) has been widely used to mitigate the bias by memorizing streamed batches (Yang et al., 2022; Chiley et al., 2019; Liao et al., 2016). To preserves the property of gradient on calibrated statistics, we follow Yang et al. and Schneider et al. to implement the EMA normalization at test time. Let $\phi$ be the composed tuple, $[\mu, \sigma]$, for mean and variance in a BN layer. $\phi_t$ denotes the running statistics at iteration $t$ and $\hat{\phi}_t$ is the one from the $t$-th batch. The EMA statistics are

$$\phi_t = (1 - \beta)\phi_{t-1} + \beta\hat{\phi}_t, \tag{4}$$

where the parameter $\beta \in [0, 1]$ governs the memory length and therefore works as the *forget gate*. A small $\beta$ endorses the model with long-term memory, otherwise short-term memory.

Traditionally, $\beta$ is constant at running time, which however cannot accommodate the estimation to the *dynamic* distributions $P_t$. Intuitively, when a model is stably running in a single domain, e.g., $P_t \approx P_t$, $\beta$ should be small to keep as many data points as possible to support accurate statistic estimation. In contrast, when the distribution is shifting, e.g., $P_t \neq P_{t-1}$, $\beta$ should be large for avoiding the mixture of two distinct statistics. Complying with the intuition, we introduce a forgetting gate to calibrate the $\beta$ adaptively, $\beta_t = h(\phi_{t-1}, \hat{\phi}_t)$, where $h(\cdot, \cdot)$ captures the distributional shifts. We consider a non-parametric heuristic definition of $h(\cdot, \cdot)$ as follows.

$$\beta_t = 1 - e^{-D(\phi_{t-1}, \hat{\phi}_t)}, \; D(\phi_{t-1}, \hat{\phi}_t) = \frac{1}{C} \sum_{i=1}^{C} KL(\phi_{t-1,i} \| \hat{\phi}_{t,i}) + KL(\hat{\phi}_{t,i} \| \phi_{t-1,i}), \tag{5}$$

where $D(\cdot, \cdot)$ is a properly-defined distance function measuring distribution shifts. The KL divergence, $KL(\phi_1^i \| \phi_2^i)$, is defined as $\log \sigma_2^i - \log \sigma_1^i + \frac{1}{2\sigma_2^{i\,2}}(\sigma_1^{i\,2} + (\mu_1 - \mu_2)^2) - \frac{1}{2}$ assuming two Gaussian distributions parameterized by $\phi_1$ and $\phi_2$, respectively. The distance function is inspired by Li et al. (2017), where the authors showed that $\phi_1$ and $\phi_2$ have larger KL divergence (based on a Gaussian assumption) if they are from distinct domains. In addition, $\beta_t^l$ is estimated layer by layer, as the distribution of different layers will shift to different degrees. Intuitively, when the $(l-1)$-th layers are well aligned after calibrating $\phi_t^l$, the deep layers should be aligned better.

**(Reduce $C$) Sparse gradients via stochastically-pruned caches.** It is easy to see that dropping the caches the channel $i$ in $z$ will vanish the corresponding gradient at the channel and may leave

the corresponding affine parameter underfitted. Thus, trivially pruning channels would cast serious issues, especially when some channels are critical for OOD generalization (Sehwag et al., 2020). But, it is hard to predict which gradient is such important not to be pruned before it is computed. Therefore an efficient pruning strategy without depending on back-propagation is desired. For this purpose, we propose an unconditioned pruning strategy by repeatedly generating a stochastic mask $M$ *per iteration* such that $q \times 100\%$ entries of the mask are zeros and the rest are ones. Given the input tensor $z$ to the affine layer, we mask the tensor by $\tilde{z}_{n,i,j,k} = M_i z_{n,i,j,k}$ for caches in Eq. (3).

Our pruning strategy has multi-fold merits. Since forwarding is not influenced, the prediction can be done still on the full size of the network preserving a high quality of semantic features. Meanwhile, pruning lowers memory usage significantly with much smaller intermediate caches and gradients. The recomputed masks per iteration resembles a progressive learning paradigm and the momentum technique in modern optimizers, like SGD or Adam, can impute the missing gradients. Moreover, the approach mitigates catastrophic forgetting since only a subset of affine weights are updated and the low-magnitude parameters are not updated. Specifically, given gradients $g_t$, the model difference $\|\theta_t - \theta_0\| = \|\sum_t g_t\| \leq \mathcal{O}(\sum_t \|g_t\|)$ will be reduced with some zeroed $\|g_t\|$, which can be viewed as an implicit anti-forgetting regularization in EATA. Later, we empirically demonstrate the regularization effect in mitigating the forgetting in regularization-free Tent.

**(Dynamic $L$) Train layers on demand.** Most test-time adaptations will last for a long time in a single environment. For example, an auto-driving car will keep running in sunny weather for long daytime, producing thousands of images. Continually adapting the model to the same environment for an overly-long time will not improve models continually but waste resources. Thus, we propose to stop the back-propagation if the optimization converges and we restart it on demand of adaptation. The principle is that *the adaptation is demanded when data distribution fundamentally shifts.* Recalling that we already measure the distributional shift by Eq. (5), we reuse the metric to guide the adaptation. Specifically, we use a threshold $\beta_{\text{th}}$ to make the decision. If $\beta_t^l \leq \theta_{\text{th}}$, then the $z^l$ is cached for back-propagation. Because of the layer-wise decision, any layer can halt training to save a lot of memory early before the network is fully executed or the optimization of all layers converge. This technique can introduce dynamic memory allocation, benefiting multi-task mobile systems. For example, the extra memory freed by the training can be used for other apps on the smartphone.

Finally, we summarize the proposed method in Algorithm 1, where our method includes three hyper-parameters to trade off accuracy and memory. Notably, our method is fully enclosed in a MECTA Norm layer, which can be easily embedded into a network, like the widely-used ResNet (He et al., 2016), to enhance the memory efficiency at test-time adaptation.

---

**Algorithm 1** Memory-Economic Continual Test-time Adaptation (MECTA)

**Input**: A model $f_\theta$ with $L$ MECTA Norm layers, the total number of iterations $T$, pruning rate $q$, batch size $B$ and layer cache threshold $\beta_{\text{th}}$.

1: **for** iteration $t \in \{1, \cdots, T\}$ **do**
2:      Initiate intermediate cache: $Z = \emptyset$
3:      **for** MECTA Norm layer $l \in 1, \ldots, L$ **do**
4:          Get layer input $x^l$;
5:          Compute the current batch statistics $\hat{\phi}_t^l$;
6:          Compute forget gate $\beta_t^l$ by Eq. (5) and the statistics $\phi_t^l$ by Eq. (4);    ▷ Stable statistics with reduced $B$
7:          Compute $a_t^l$ and $z^l$ by Eq. (2) using $\phi_t^l$;
8:          **if** $\beta_t^l > \beta_{\text{th}}$ **then**                                      ▷ Dynamic $L$
9:             Randomly remove $q \times 100\%$ of channels in $z_t^l$ and cache $Z = Z \cup \{z_t^l\}$;         ▷ Reduce $C$
10:        Output $a^l$ to the next layer;
11:      Compute loss and back-propagate gradients with cache $Z$ to update parameters;

---

**Critical difference to prior work on memory footprint reduction.** Previous attempts to reduce footprint (Wang et al., 2022b; Yuan et al., 2021) focused on parameter sparsity and reduce the overhead of storing model parameters and gradients. However, the overhead of parameters or gradients is relatively small in comparison to the caches for backwarding, as shown in Fig. 1. Instead, we focus on the large overheads of caches and our method is ready to work with the traditional parameter sparsity. For this purpose, our channel pruning and on-demand layer training resembles the gradient sparsity (Garg & Khandekar, 2009) or coordinate descent (Wright, 2015), and the two tactics implicitly prune the gradients before back-propagation avoiding large caches. Other than gradient

sparsity, gradient checkpointing (GC) Chen et al. (2016) is a more general way to reduce memory footprint. GC only caches the inputs of segments of layers and recompute intermediate features on demand. The optimal memory reduction is about the $1/\sqrt{L}$ of the original cost. Both GC and our method can greatly reduce the memory, but our method is favored because MECTA can reduce the memory on demand, MECTA is more computation efficient.

## 5 EXPERIMENTS

**Datasets and pre-trained models.** To evaluate the OOD generalization of models, we adopt three image-classification datasets: the CIFAR10-C, CIFAR100-C (Krizhevsky, 2009) and ImageNet-C (Deng et al., 2009) following previous arts (Niu et al., 2022). All the datasets are processed with 15 kinds of corruptions from 4 main categories (noise, blur, weather, and digital ones) with the highest severity level 5, which are widely used as the robustness benchmark for deep learning (Hendrycks & Dietterich, 2019). Without specifications, we use the ResNeXt29-32×4d pre-trained by Aug-Mix (Hendrycks et al., 2020b) for CIFAR10-C and CIFAR100-C, and the ResNet50 pre-trained by Hendrycks et al. (2020a) for ImageNet-C, which has compelling robust performance in the benchmark learderboard (Croce et al., 2021).

**Baselines.** Our method is ready to replace batch-normalization in adaptable networks for better OOD generalization. **(1) CTA.** Therefore, we plug the proposed MECTA into two backbone methods: Efficient Anti-forgetting Test-time Adaptation (EATA) (Niu et al., 2022) and Test-time entropy minimization (Tent) (Sun et al., 2020). We also include the simplest BN statistic adaptation (BN) (Schneider et al., 2020) which is the most memory- and computation-efficient test-time adaptation only updating BN statistics without gradients. Tent improves the model accuracy via self-supervision and constrains the parameter updating to affine layers. EATA is the state-of-the-art efficient solution that balances computation efficiency and accuracy well. We plug MECTA into them to show the potential improvements in accuracy, efficiency, and generality on methods with advanced principles. Though other methods are also applicable to the problem, they typically lack advantages in efficiency, thus out of our scope. **(2) Memory reduction.** Gradient checkpointing (GC) Chen et al. (2016) is a general method for reducing memory costs in training, which trades in computation for memory and is applicable with any sequential structure. We apply GC to ResNet where we treat each block (including several conv-bn-relu layers) as the minimal unit and segment ResNet into $\sqrt{L}$ parts.

**Evaluation setup.** We conduct experiments on the *lifelong* setting, where the batches of data come in a streaming manner. For privacy considerations, no data shall be stored for memorization. Similarly, the types of corruption also arrive sequentially and are agnostic to the model. Thus, models have to be adapted online to gain higher accuracy on streaming data and environments. Such a setup can simulate a challenging yet realistic scenario, admitting high practical values.

**Evaluation metrics.** We adopt two metrics to evaluate the effectiveness and memory efficiency of different methods. (1) *Accuracy* (%) is evaluated on all samples under the same corruption with the highest level 5 by default. (2) *Cache size* (Mb). Since the memory of model parameters and optimizers is always constant or linearly dependent on the size of intermediate variables, we focus on the tensor size of intermediate variables. Specifically, we sum up the memory consumption of the tensor for affine layers, i.e. the *cache* $z^l$ in Eq. (3). For the ease of memory accountant, we eclude other memory costs that are the same among different algorithms.

More details regarding hyper-parameters and implementation can be found in the appendix.

### 5.1 BENCHMARKS ON OOD PERFORMANCE

**Comparison under the same cache constraint.** We evaluate the accuracy with a constraint on the cache size. For this purpose, we let the gradient-free methods use a large batch size of 128 and a gradient-based method (Tent and EATA) select a small batch size from $\{4, 8, 16, 32\}$ such that the latter has a smaller maximal cache size than the former does. In Table 1, we report the per-domain accuracy in continual test-time adaptation in the column order, and we calculate the cache size on average and on maximum along the sequential adaptation. We have two main observations on MECTA. **(1) MECTA improves accuracy under cache constraints.** With the constraint, Tent and EATA have to adopt a small batch size and therefore perform poorly, even worse than gradient-free adaptation (BN). By reducing the cache dynamically, MECTA boosts the performance of EATA and Tent drastically by $6-71.6\%$ on all three datasets. EATA+MECTA shows the best performance in CIFAR10-C and ImageNet-C and comparable accuracy to Tent on CIFAR100-C. **(2) MECTA is more computation- and memory-efficient than GC.** Though GC lowers the memory costs,

it meanwhile increases the computation costs by around $20\%$ in terms of GFLOPs. In contrast, our method only marginally increases the computation cost by less than $0.2\%$ and can bear larger batches.

Table 1: Continual evaluation on three datasets regarding accuracy ($\%$) and cache sizes (Mb). For a fair comparison, batch sizes (BS) are chosen such that the corresponding cache sizes are lower than those of BN with a batch size of $128$. Orig. denotes the original data. Blue cells highlight the highest accuracy on the same dataset, and the bold texts indicate the best accuracy given the same base algorithm. GFLOPs is the number of $10^9$ Floating Point Operations for adapting one sample.

| Alg. | BS | Noise Gauss. | Shot. | Impul. | Blur Defoc. | Glass. | Motion | Zoom. | Weather Snow | Frost | Fog | Bright. | Digital Contr. | Elast. | Pixel. | JPEG | Orig. | Acc. Avg | Cache Avg | Max | GFLOPs |
|---|---|---|---|---|---|---|---|---|---|---|---|---|---|---|---|---|---|---|---|---|---|
| *CIFAR10-C* | | | | | | | | | | | | | | | | | | | | | |
| BN | 128 | 81.5 | 83.2 | 79.9 | 92.4 | 80.7 | 91.1 | 92.1 | 87.9 | 87.7 | 85.7 | 92.7 | 89.8 | 85.6 | 87.6 | 83.0 | 94.2 | 87.2 | 134 | 134 | 1.1 |
| Tent | 8 | 68.8 | 46.3 | 16.4 | 12.8 | 11.8 | 8.9 | 9.9 | 10.7 | 10.3 | 10.2 | 10.3 | 10.2 | 10.1 | 10.0 | 10.2 | 10.2 | 16.7 | 114 | 114 | 2.180 |
| +GC | 16 | 81.8 | 82.2 | 71.8 | 68.9 | 52.8 | 43.7 | 39.0 | 30.7 | 22.8 | 17.7 | 10.1 | 6.7 | 7.4 | 8.2 | 8.1 | 9.0 | 35.1 | 130 | 130 | 2.6 |
| +MECTA | 31 | **86.5** | **87.2** | **81.3** | **88.3** | **78.9** | **84.1** | **85.8** | **81.3** | **80.8** | **77.4** | **82.1** | **77.4** | **74.9** | **76.6** | **71.7** | **77.5** | **80.7** | 93 | 129 | 2.182 |
| EATA | 8 | 74.5 | 70.9 | 66.6 | 69.7 | 55.5 | 54.6 | 47.2 | 38.4 | 33.1 | 35.1 | 35.2 | 18.8 | 12.2 | 14.5 | 13.3 | 8.5 | 40.5 | 114 | 114 | 2.180 |
| +GC | 16 | 83.4 | 84.3 | 81.3 | 86.4 | 77.0 | 83.8 | 86.5 | 84.5 | 84.7 | 83.9 | 88.1 | 87.9 | 80.9 | 85.5 | 79.3 | 87.9 | 84.1 | 130 | 130 | 2.6 |
| +MECTA | 31 | **86.6** | **88.3** | **84.4** | **89.2** | **82.4** | **87.4** | **89.3** | **86.5** | **87.6** | **86.4** | **89.6** | **88.0** | **85.1** | **87.7** | **83.1** | **90.3** | **87.0** | 102 | 130 | 2.182 |
| *CIFAR100-C* | | | | | | | | | | | | | | | | | | | | | |
| BN | 128 | 57.6 | 59.0 | 56.6 | 72.5 | 58.2 | 69.9 | 71.8 | 64.7 | 64.8 | 57.9 | 73.5 | 69.8 | 64.3 | 66.7 | 58.6 | 75.8 | 65.1 | 134 | 134 | 1.1 |
| Tent | 8 | 52.9 | 53.9 | 46.7 | 50.5 | 31.2 | 29.7 | 23.7 | 14.3 | 10.0 | 6.7 | 5.9 | 3.4 | 3.9 | 3.6 | 3.5 | 3.5 | 21.5 | 114 | 114 | 2.180 |
| +GC | 16 | 56.8 | 61.4 | 59.6 | 68.5 | 56.1 | 65.3 | 66.9 | 59.7 | 60.0 | 54.9 | 65.0 | 57.0 | 54.2 | 56.2 | 46.8 | 58.6 | 59.2 | 130 | 130 | 2.6 |
| +MECTA | 31 | **58.8** | 61.2 | 58.2 | **73.2** | **60.7** | **71.4** | **73.4** | **65.8** | **66.7** | **60.1** | **73.6** | **68.4** | **65.5** | **67.3** | **59.1** | **75.1** | **66.2** | 77 | 130 | 2.182 |
| EATA | 8 | 52.1 | 54.2 | 53.2 | 65.3 | 51.5 | 63.8 | 64.9 | 59.1 | 58.5 | 53.9 | 66.8 | 63.2 | 56.3 | 61.1 | 52.9 | 70.1 | 59.2 | 114 | 114 | 2.180 |
| +GC | 16 | 57.3 | 60.5 | **58.5** | 69.9 | 57.1 | 68.9 | 69.8 | 63.7 | 64.4 | 59.4 | 71.6 | 67.9 | 62.8 | 67.1 | 58.2 | 74.4 | 64.5 | 130 | 130 | 2.6 |
| +MECTA | 31 | **58.7** | **60.7** | 57.8 | **72.7** | **59.4** | **70.9** | **73.6** | **65.8** | **66.8** | **60.4** | **74.7** | **70.7** | **66.0** | **68.4** | **60.6** | **77.7** | **66.6** | 75 | 130 | 2.182 |
| *ImageNet-C* | | | | | | | | | | | | | | | | | | | | | |
| BN | 128 | 39.2 | 42.6 | 39.6 | 29.9 | 32.9 | 40.8 | 47.4 | 45.0 | 47.7 | 55.8 | 68.5 | 36.0 | 54.8 | 65.4 | 55.7 | 74.2 | 48.5 | 411 | 411 | 4.1 |
| TENT | 8 | 33.8 | 16.5 | 0.8 | 0.4 | 0.3 | 0.4 | 0.4 | 0.4 | 0.4 | 0.3 | 0.4 | 0.2 | 0.3 | 0.4 | 0.3 | 0.4 | 3.5 | 355 | 355 | 8.183 |
| +GC | 16 | 43.3 | 46.1 | 42.8 | 25.8 | 14.8 | 5.0 | 1.3 | 0.7 | 0.7 | 0.7 | 0.8 | 0.6 | 0.7 | 0.7 | 0.7 | 0.7 | 11.6 | 404 | 404 | 10.3 |
| +MECTA | 30 | **48.6** | **50.9** | **48.5** | **35.7** | **38.3** | 39.6 | **44.2** | **37.0** | **37.4** | **42.1** | **51.9** | **31.7** | **42.9** | **47.6** | **42.5** | **53.6** | **43.3** | 338 | 397 | 8.190 |
| EATA | 8 | 34.1 | 37.0 | 35.0 | 27.5 | 28.1 | 35.5 | 38.6 | 39.6 | 39.7 | 47.8 | 56.6 | 36.5 | 44.1 | 53.3 | 46.7 | 63.2 | 41.4 | 355 | 355 | 8.183 |
| +GC | 16 | 44.4 | 47.1 | 45.4 | 39.0 | 39.4 | 47.4 | 49.7 | 49.7 | 48.4 | 57.6 | 64.3 | 47.8 | 54.5 | 61.7 | 56.3 | 69.5 | 51.4 | 404 | 404 | 10.3 |
| +MECTA | 30 | **50.6** | **53.3** | **51.7** | **44.7** | **46.1** | **52.2** | **56.1** | **53.4** | **53.0** | **62.0** | **68.9** | **52.9** | **60.4** | **67.1** | **61.7** | **73.6** | **56.7** | 342 | 397 | 8.190 |

**Comparison using the same batch size.** In Table 2, we follow the CTA protocol of (Niu et al., 2022) to compare the proposed method with other baselines, e.g., TTT (Sun et al., 2020), TTA (Ashukha et al., 2021), MEMO (Zhang et al., 2021) and CoTTA (Wang et al., 2022a). For a fair comparison, all methods only adapt BN layers if applicable, and use the standardly-trained ResNet50, which is publicly available, for example, from the official PyTorch package. **(1) MECTA is memory-efficient with the same batch size.** With only $30\%$ of cache (or $52\%$ of GC cache), MECTA maintains similar performance as the standard EATA. Notice that GC is less effective in cache reduction here than the reported results in (Chen et al., 2016), because the parameter-efficient adaption already drops a lot of caches for frozen convolutional layers. The cache cost of EATA+MECTA is pretty low which is only more than those of the BN and non-adapted source models, yet has a large gain ($12\%$) on accuracy meanwhile. **(2) MECTA mitigates forgetting.** Note Tent has been known to suffer from catastrophic forgetting in continual adaptation (Niu et al., 2022) and thus has decreasing trend on the accuracy by sequential domains, and we find that our method can mitigate this issue. The inherent reason for the mitigation is quite intuitive: the MECTA will adaptively reduce the number of parameters to be updated, which is similar to the regularization of EATA on norm-bounded parameter distance. Also, we add the original dataset as the last domain in CTA to evaluate if the model forgets the original data. Both Tent and EATA have significantly lower accuracy on the original set with small batch size in Table 1, while Tent/EATA+MECTA gains much higher original-set accuracy on all datasets.

## 5.2 QUALITATIVE STUDIES

In the following experiments, if not specified, we use ImageNet-C and robustly pre-trained ResNet50.

**Effect of each component in ECTA on memory-accuracy trade-off.** For brevity, we name the components as MECTA-X where X is **B** for adaptive forgetting, **C** for stochastic cache pruning and **L** for adaptive layer training. We define the accuracy averaged over all corruptions as the *mean-corruption acc*. An overall comparison with EATA and BN on mean-corruption acc is depicted in Fig. 2a. For BN, we vary the batch size from {32, 64, 128, 256, 512}. For EATA and EATA+MECTA-B, we vary the batch size from {4, 8, 16, 32, 64} where 64 is the default setting from

Table 2: Comparison to other baselines with a batch size of 64 on the standardly-trained ResNet50 on ImageNet-C. #fwd and #bwd denote the times of forward and backward over a batch on average. GN is the group norm and JT denotes the model is jointly trained via supervised cross-entropy and rotation prediction losses. The bold numbers indicate the highest accuracy (or smallest cache) and the underlined ones are the second best result. Cache sizes with GC are in brackets.

| | Noise | | | Blur | | | | Weather | | | | Digital | | | | Acc. | Cache | #fwd | #bwd |
|---|---|---|---|---|---|---|---|---|---|---|---|---|---|---|---|---|---|---|---|
| Alg. | Gauss. | Shot. | Impul. | Defoc. | Glass. | Motion | Zoom. | Snow | Frost | Fog | Bright. | Contr. | Elast. | Pixel. | JPEG | Avg | Max (Mb) | | |
| *ResNet50; Reset model per perturbation* | | | | | | | | | | | | | | | | | | | |
| TTT (GN+JT) | 31.0 | 33.6 | 33.4 | 28.1 | 7.8 | 33.2 | 36.8 | 40.9 | 19.0 | 51.0 | 61.8 | 38.9 | 49.4 | 51.7 | 48.0 | 37.6 | 2460×20 | 21 | 20 |
| BN | 15.5 | 16.1 | 16.3 | 20.0 | 20.0 | 28.5 | 40.0 | 34.8 | 35.0 | 48.5 | 65.9 | 24.1 | 45.8 | 50.7 | 41.1 | 33.5 | **206** | 1 | 0 |
| TTA | 4.1 | 4.9 | 4.5 | 12.5 | 8.2 | 12.9 | 25.8 | 14.0 | 19.1 | 21.3 | 53.0 | 12.4 | 14.6 | 24.6 | 33.6 | 17.7 | **206** | 64 | 0 |
| MEMO | 7.5 | 8.7 | 9.0 | 19.7 | 13.0 | 20.7 | 27.6 | 25.3 | 28.8 | 32.1 | 61.0 | 11.0 | 23.8 | 33.0 | 37.5 | 23.9 | 2460×65 | 65 | 65 |
| *ResNet50; Lifelong adaptation* | | | | | | | | | | | | | | | | | | | |
| CoTTA(+GC) | 16.9 | 20.3 | 22.8 | 20.6 | 22.0 | 31.7 | 42.4 | 34.5 | 34.0 | 47.2 | 58.9 | 24.1 | 44.5 | 48.6 | 42.4 | 34.1 | 2845 (1618) | 33 | 1 |
| Tent(+GC) | 28.4 | 34.1 | 31.7 | 19.3 | 12.2 | 6.9 | 4.0 | 1.4 | 0.8 | 0.7 | 0.9 | 0.4 | 0.6 | 0.7 | 0.6 | 9.5 | 2845 (1618) | 1 | 1 |
| Tent+MECTA | 24.5 | 29.5 | 28.3 | 22.0 | 23.5 | 27.4 | 37.2 | 28.2 | 27.1 | 36.8 | 50.7 | 15.5 | 38.0 | 40.2 | 34.7 | 30.9 | 847 | 1 | 1 |
| EATA(+GC) | **35.0** | 38.1 | 36.8 | 33.8 | 34.2 | 47.3 | 53.2 | 51.1 | 45.6 | 59.7 | 68.0 | 44.2 | 57.2 | 60.4 | 54.7 | 48.0 | 2845 (1618) | 1 | **0.56** |
| EATA+MECTA | 33.7 | **39.1** | **37.8** | 31.7 | 33.1 | 42.2 | 50.3 | 46.3 | 43.0 | 56.9 | 65.4 | 41.2 | 55.2 | 58.2 | 53.7 | 45.9 | 847 | 1 | **0.56** |

the original paper of EATA. For EATA+MECTA-BL, we let the batch size be 16 and vary the $\beta_{th}$ in $\{1, 6.25, 12.5, 25, 50, 100, 200, 400\} \times 10^{-4}$. For EATA+MECTA-BLC, we let the batch size be 16 and $\beta_{th}$ be 0.00125 and vary the pruning rate $q$ in $\{0.1, 0.3, 0.5, 0.7, 0.9\}$.

The simplest BN method reaches the best accuracy at the batch size of 128 with approximately 400 Mb cache, which is much lower than the best accuracy achieved by EATA. However, given the same cache size, EATA does not perform better than BN. Only when 1200 Mb cache memory is occupied, EATA significantly outperforms BN by 5%. The need for a large batch (corresponding to a large cache) by EATA comes from the poor performance when the batch size is small, e.g., at a batch size of 8 or 16. MECTA-B treats EATA with a streaming normalization with adaptive forgetting and memorization leading to higher accuracy on small-batch adaptation. EATA+MECTA-B achieves better or comparable accuracy using either the smallest memory or the largest memory. Meanwhile, MECTA-L and MECTA-C improve the trade-off frontier into a low-cost zone. Given cache size lower than 600 Mb, MECTA-BL can beat the best accuracy of EATA. Last, with cache pruning, MECTA-BLC further improves the efficiency more impressively: best accuracy at an extremely small cache size (fewer than 100 Mb).

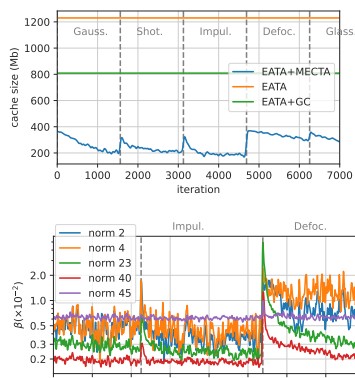

Figure 3: Dynamic cache sizes and layer-wise $\beta$ using MECTA.

To understand the cache reduction, we study the memory/accuracy by three MECTA components.
**(L) Adaptive layer-training.** In Fig. 3, we show that the cache size will dynamically change by iterations using EATA+MECTA (batch size of 32). Especially, the cache size will peak at the beginning of distribution shifts and gradually vanishes. The $\beta$ differs by layer and different layer has a different sensitivity to the distributional shifts, which motivate us to use set $\beta$ and activate training layer-wisely. In Fig. 2b, when $\beta_{th}$ is small, all layers will be trained and therefore the cache size is larger than the efficient BN adaptation. Increasing $\beta_{th}$ reduces the cache size on average of the adaptation process. Though this meanwhile decreases the accuracy, the accuracy is still higher than the BN adaptation by more than 4% based on the same batch size.

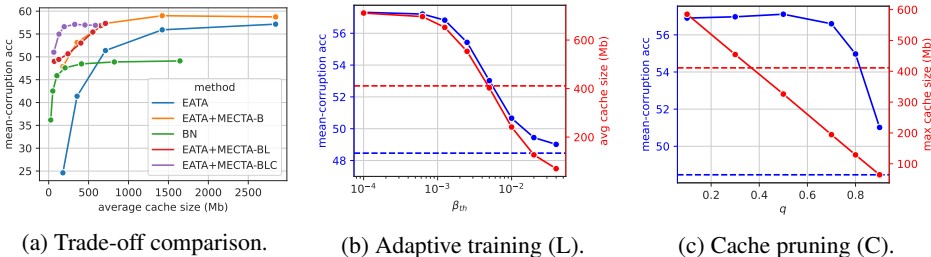

(a) Trade-off comparison.     (b) Adaptive training (L).     (c) Cache pruning (C).

Figure 2: Ablation study on the components of MECTA. The dash horizontal lines in (b) and (c) are the performance of BN with the best trade-off in (a) (of batch size 128).

Table 3: Ablation study of MECTA-B regarding accuracy (%) and a batch size of 16. Bold texts indicate the best accuracy among ablations. See Table 7 for other backbone methods.

| | EMA | MECTA-B | Noise | | | Blur | | | | Weather | | | Digital | | | | | |
|---|---|---|---|---|---|---|---|---|---|---|---|---|---|---|---|---|---|---|
| Alg. | $\beta = 0.1$ | auto $\beta$ | Gauss. | Shot. | Impul. | Defoc. | Glass. | Motion | Zoom. | Snow | Frost | Fog | Bright. | Contr. | Elast. | Pixel. | JPEG | Orig. | Avg |
| | ✗ | ✗ | 44.4 | 47.1 | 45.4 | 39.0 | 39.4 | 47.4 | 49.7 | 49.7 | 48.4 | 57.6 | 64.3 | 47.8 | 54.5 | 61.7 | 56.3 | 69.5 | 51.4 |
| EATA | ✓ | ✗ | 49.2 | 52.0 | 50.6 | 43.8 | 45.1 | 52.5 | 55.5 | 54.4 | 53.5 | 62.3 | 69.6 | 52.1 | 60.0 | 67.4 | 61.6 | 74.4 | 56.5 |
| | ✓ | ✓ | **50.0** | **53.1** | **51.5** | **44.8** | **45.7** | **53.2** | **56.5** | **55.3** | **54.3** | **63.2** | **70.3** | **52.6** | **61.1** | **68.1** | **62.5** | **75.0** | **57.3** |

Table 4: Evaluation of $k$-new $K$-old shift accuracy by EATA. Average accuracy (AA %) and worst accuracy (WA %) are reported for each target perturbation. Values in the brackets denote the difference between the current method and the base one. More combinations of $(K, k)$ are in Table 8.

| $K$ | $k$ | EMA | MECTA-B | Impul. | | Motion | | Fog | | Elast. | |
|---|---|---|---|---|---|---|---|---|---|---|---|
| | | $\beta = 0.1$ | Auto $\beta$ | AA | WA | AA | WA | AA | WA | AA | WA |
| | | ✗ | ✗ | 35.5 | 34.6 | 37.0 | 36.8 | 50.1 | 49.8 | 48.3 | 47.8 |
| 49 | 1 | ✓ | ✗ | 35.4 (**-0.1**) | 30.6 (-4.0) | 26.6 (-11.0) | 20.1 (-16.7) | 41.0 (-9.1) | 25.9 (-23.9) | 42.8 (**-5.5**) | 39.3 (-8.5) |
| | | ✓ | ✓ | 34.4 (-1.1) | 32.0 (**-2.6**) | 28.6 (**-8.4**) | 25.7 (**-11.1**) | 43.6 (**-6.5**) | 39.1 (**-10.7**) | 42.6 (-5.7) | 41.2 (**-6.6**) |

**(C) Channel pruning.** Fig. 2c shows that the pruning linearly reduces the maximal cache size, which bounds the maximal memory requirement in the whole adaptation life-cycle. With such a reduction in memory, pruning only mildly sacrifices the accuracy by less than 1%.

**(B) Adaptive forgetting.** In Fig. 2a, we observe that small batch size (corresponding to small cache size) hurts the accuracy of EATA significantly. Thus, it is crucial to ask *how the proposed adaptive forgetting and memorization improve CTA accuracy in small batches.* To answer this question, we reiterate the two keys of CTA: accurate modeling of data distribution from stream data and stability in dynamic environments. We show that our adaptive forget gate can achieve the two goals simultaneously without hyper-parameters compared to the traditional moving average method.

**(B.1) Parameter-free memorization improves the per-domain accuracy.** In Table 3, we present the per-domain accuracy on the full perturbation set with batch size 16 using all perturbations. We include the exponential moving average (EMA) to show that memorizing streaming batches into a moving average can improve the accuracy of small batch sizes. Despite the effectiveness of EMA, the hyperparameter $\beta$ requires extra effort or extra labeled data in tuning. In contrast, MECTA-B is parameter-free and meanwhile outperforms the EMA on all corrupted and original data.

**(B.2) Forgetting improves shift stability.** Regarding the advantage of adaptive forgetting in Table 3, we conjecture that the adaptive $\beta_t$ estimated by MECTA-B can benefit the accuracy when the data distribution shifts from one domain to another, namely, *shift accuracy* (stability). To quantify the shift accuracy, we suppose that a model first experiences $K$ batches from one domain (old domain), and then receives $k$ batch from another domain (new domain). The shift accuracy is the correct ratio of model predictions on samples of the $k$-new batches. For example, we measure the shift accuracy by randomly sampling 49 batches from the old domain and retrieving one batch from the new domain without replacement until all data of the new domain have been evaluated. For a given kind of perturbation as a new domain, we evaluate the shift accuracy paired with all kinds of perturbation as the old domain and report the *average-case accuracy* (AA) and the *worst-case accuracy* (WA). Here, we use ResNet50 on ImageNet and a *subset of perturbations* for the ease of intensive evaluations: Impul, Motion, Fog, and Elast. As summarized in Table 4, EMA causes large accuracy declines, compared to the batch-estimated BN statistics (without distribution memory). This observation implies that in contrast to the large gains in continual evaluation, the distribution memory induced by EMA becomes an obstacle for models to efficiently adapt new domains. Instead, short-term memory is preferred. Without explicitly switching a hyper-parameter, our method automatically detects such domain shift and merits the accuracy on shift with memory from past domains.

## 6 CONCLUSION

Though the gradient-based Continual Test-time Adaptation (CTA) is an effective manner to improve running performance on test data, the high memory consumption becomes a considerable challenge for deploying the technique to resource-limited end devices, like smartphones. To our best knowledge, our work is the pilot study on the memory efficiency and reveal the main bottleneck is on the intermediate results cached for back-propagation, even if only few parameters need to be updated in the state-of-the-art computation-efficient solution (EATA). Therefore, we propose the first cache-aware method that achieves higher accuracy with much lower cache memory.

## ETHICS STATEMENT

In this paper, we work is not related to human subjects, practices to dataset releases, discrimination/biases/fairness concerns, adn also do not have legal compliance or research integrity issues. Our work is proposed to address the memory efficiency of model adaptation in continual distributional shifts. In this case, if the model is trained out of good will, we have good faith that our methods won't cause any ethic problem or negative social impacts.

## REPRODUCIBILITY STATEMENT

The algorithm pseudo codes are enclosed in the main body. We also provide details on the implementation, hyper-parameters in Appendix B.1. The datasets and baselines codes are all public available online and we specify them properly in our paper.

### ACKNOWLEDGMENTS

This research was funded by Sony AI. This material is based in part upon work supported by the National Science Foundation under Grant IIS-2212174, IIS-1749940, ECCS-2024270, Office of Naval Research N00014-20-1-2382, and National Institute on Aging (NIA) RF1AG072449.

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

## A DISCUSSION

### A.1 MEMORY FOOTPRINT FOR TEST-TIME ADAPTATION

When a learning algorithm looks for deployment on edge devices, it is unavoidable to take the two folds into account: (1) whether the maximal memory consumption of the algorithm can be fitted into a mobile device; (2) whether the dynamic and instantly-free space of device memory (RAM) is sufficient for the adaptation. The current mobile devices generally have memory in GB levels. For example, current general mobile devices such as Samsung S20 have 2GB or more memory, Raspberry Pi has 1GB or more memory. However, unlike the training or adaptation on a high-capacity GPU server where all the resources (memory and CPU time) can be allocated for training, the memory on mobile devices may be temporarily and partially allocated for the operating system and other background applications. The situation could be even more severe for mobile devices. Thus, squeezing the memory footprint dynamically is crucial for edge devices.

To avoid losing focus of the paper, we only consider the cache for gradient computation on batch-normalization layers, which has been a substantially-larger memory footprint compared to those of the widely-studied model and gradient memory in the literature (Yuan et al., 2021; Akin et al., 2019; Wang et al., 2022b). Admittedly, our work does not cover all memory consumption in the life-span of model adaptation. In Table 1, we only compute the memory costs (cache) of the back-propagation but not the forward operations, because the memory costs of the latter will be released immediately. The actual memory occupation in hardware, like NVIDIA GPU, will be enlarged due to the reservation for faster inference. The holistic solution that jointly optimizes the inference time and memory requires extra efforts on low-level computation and could be hardware-dependent. Thus, we leave the solution as a open problem for future study.

## B EXPERIMENTAL DETAILS AND SUPPLEMENTARY

### B.1 IMPLEMENTATION DETAILS

**Hyper-parameters.** All test-time adaptation objectives are optimized by stochastic gradient descent (SGD) with a momentum of 0.9. Tent and EATA utilize a batch size of 64 with a learning rate of 0.005 (0.00025) for CIFAR-10 (CIFAR100 and ImageNet). In our implementation, we use 0.0025 (0.0001) as learning rates to stabilize the training with smaller batch sizes. EATA uses 2,000 samples to estimate a Fisher matrix for anti-forgetting regularization. For MECTA, we set the threshold $\beta_{th}$ for stopping layer training as 0.0025 for CIFAR100, 0.00125 for CIFAR10, and 0.00125 for ImageNet-C. The cache pruning rate is set to be 0.7 for all datasets.

We implement our algorithm using PyTorch 1.12.1, cudatoolkit 11.6 on NVIDIA Tesla T4 GPUs. The codes of baselines are provided by the open sourced codes of EATA[1]. For gradient checkpointing, we use the official implementation from PyTorch, `torch.utils.checkpoint`. For each stage in the ResNet with $m$ blocks, we will split the blocks sequentially into $\lfloor m/2 \rfloor$ segments. Therefore, ResNet50 will be split into 7 segments, approximatedly equal to $\sqrt{50}$.

**Measuring cache sizes.** Without GC, we measure the cache size by summing up the tensor size of all features $z^l$ in a network. With GC, we estimate the cache size as two parts: one is the segment cache sizes and the other is the maximal cache for backwarding inside a segment. In ablation of $\beta$ and evaluation of shift accuracy, we ignore the BN's in shortcut layers (or downsample layers) in variants of ResNet and use a simplified implementation.

**Measuring full footprint.** In Fig. 1, we track the tensors that are cached in the GPU memory using the public tool[2]. The memory tracker will find all the PyThorch tensor variables in the garbage collector of Python. We also leverage tool, `torch.cuda.memory_cached`, provided by PyTorch to estimate the maximal GPU memory costs including non-tensor variables.

---

[1] https://github.com/mr-eggplant/EATA
[2] https://github.com/Oldpan/Pytorch-Memory-Utils

**Implementation of stochastic cache.** During forward, we will only store the remained values of $z^l$ and the indexes of the remained channels (denoted as $R$). Later, we compute the gradient by

$$\sum_{n=1}^{B} \frac{\partial \ell_n}{\partial \gamma_i^l} = \begin{cases} \sum_{n=1}^{B} \sum_{j=1}^{W} \sum_{k=1}^{H} \frac{\partial \ell_n}{\partial a_{i,j,k}^l} z_{n,i,j,k}^l, & i \in R, \\ 0, & i \notin R. \end{cases}$$

As the zero values do not need cache, the implementation can effectively reduce memory consumption with a small extra space for storing the index set $R$.

## B.2 MORE EXPERIMENTAL RESULTS

**MECTA in different network architectures.** It is known that larger and deeper models will merit the robustness of neural networks (Hendrycks et al., 2020a). In Fig. 4, we compare the EATA and EATA+MECTA on varying model depths. We adopt the protocol in Table 2 but reduce the batch size to 32 to accommodate the huge memory cost of deeper networks. The experiments are conducted with 4 perturbations. By increasing the depth of ResNet, the cache size increases steeply. Instead, MECTA reduces the cache consumption to a low level and achieves even better accuracy. Beyond ResNet, we evaluate more network architectures in Table 5. For example, MobileNet (Howard et al., 2017) is designed for edge devices with limited computation resources. Since our method only modify the batch-normalization layers, it can be easily plugged into these networks. Except wide ResNet (WRN), our method outperforms EATA with much lower cache sizes and higher accuracy.

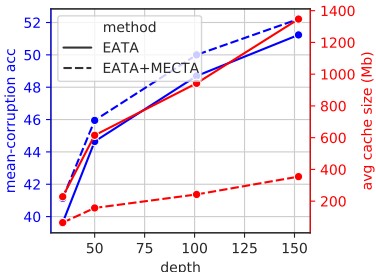

Figure 4: Evaluation on different ResNet models with varying depth.

Table 5: Evaluation on different model architectures retrived from PyTorch pre-trained models.

| Architecture | Alg. | Acc. (%) Avg | Cache (Mb) Avg | Max |
|---|---|---|---|---|
| MobileNetV2 (Howard et al., 2017) | EATA | 26.8 | 854.8 | 854.8 |
| | EATA+MECTA | **28.0** | **233.5** | **250.5** |
| MobileNetV3 (Howard et al., 2017) | EATA | 30.8 | 563.2 | 563.2 |
| | EATA+MECTA | **32.2** | **158.1** | **163.7** |
| VGG19+BN (Simonyan & Zisserman, 2015) | EATA | 34.9 | 1901.1 | 1901.1 |
| | EATA+MECTA | **35.7** | **544.8** | **564.9** |
| WRN101×2 (Zagoruyko & Komodakis, 2017) | EATA | 54.7 | 2716.7 | 2716.7 |
| | EATA+MECTA | 53.2 | **803.2** | **810.4** |
| ResNet101 | EATA | 41.2 | 1885.0 | 1885.0 |
| | EATA+MECTA | **51.3** | **449.1** | **569.8** |
| ResNet152 | EATA | 44.0 | 2694.3 | 2694.3 |
| | EATA+MECTA | **52.9** | **635.6** | **813.6** |
| ResNeXt101 32×8d (Xie et al., 2017) | EATA | 53.9 | 3802.1 | 3802.1 |
| | EATA+MECTA | **54.5** | **1094.7** | **1136.1** |
| DenseNet121 (Huang et al., 2018) | EATA | **45.9** | 2005.4 | 2005.4 |
| | EATA+MECTA | 42.2 | **594.3** | **597.6** |
| EfficientNetV2-S (Tan & Le, 2021) | EATA | 45.9 | 1556.3 | 1556.3 |
| | EATA+MECTA | **47.1** | **454.4** | **463.3** |

**Does layer-sparse training help?** In Fig. 5, we evaluate EATA with fewer trainable layers, which can reduce the cache size, following the protocol in Table 2. During adaptation, we keep $k$ deepest layers to be trained and freeze other layers, which is denoted as L$k$ in the figure. We also include the EATA+GC where we use gradient checkpointing for the trainable layers. We observe that reducing the trainable layers can significantly decrease the cache size which is even lower than MECTA. However, the corresponding accuracy is significantly decreased by 5% compared to EATA+MECTA meanwhile. In comparison, though MECTA also uses the layer-sparse training, our method presents the best accuracy-memory trade-off in the experiment. The key difference is that our method sparsifies the training only on demand, specifically when a layer is well adapted without need for further training.

**Does MECTA-B works with BN adaptation and Tent?** We show that MECTA-B can generally works well with BN adaptation and Tent, in Table 7. Consistent on all three backbone methods, the MECTA-B can outperforms EMA and base methods without extra hyper-parameters. EMA and MECTA-B can salvage BN and Tent from pooer performance using small batch sizes.

**More shift-accuracy evaluation.** We consider more cases of $(K, k)$ pairs in Table 8. In all three trials, MECTA-B reduce the shift-accuracy drops w.r.t. the baseline. Given more new-domain samples, e.g., $k = 5$, the shift-accuracy using EMA and MECTA-B becomes higher than the baseline, implying the quick convergence of the adaptation.

**Cache size and $\beta$ by iteration.** We extend Fig. 3 to a full life-cycle version in Fig. 6. For evaluating $\beta$, we run the experiment on ImageNet-C using ResNet50 and we use MECTA with EATA for adaptation. In more corruptions, we find the periodic fluctuation of $\beta$ by the distributional shifts, which results in the dynamic cache sizes.

**Does MECTA works on even smaller batches?** In Table 6, we extend Table 1 with more batch sizes. One interesting observation is that our method outperforms or is comparable to other baselines given even smaller caches. For example, given a batch size of 16, EATA+MECTA can outperform BN at the best batch size, when MECTA reduces the cache size to 71 Mb on average compared to the 134 Mb by BN.

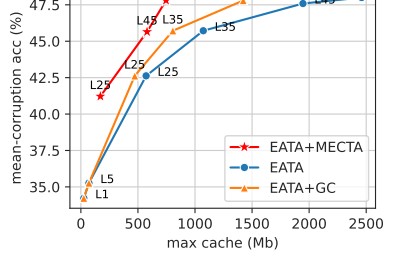

Figure 5: Compare MECTA to layer-sparse training using ResNet50 and batch size of 64. L1 means that only the deepest 1 layer is trained. Likewise, L45 denotes the deepest 45 layers.

Table 6: Continual evaluation on three datasets with the highest severity level 5 regarding accuracy (%). For a fair comparison, batch sizes (BS) are chosen such that the corresponding cache sizes are lower than those of BN with batch size of 128. Blue cells highlight the accuracy that is the highest among all methods, and the bold texts indicate the best accuracy given the same base algorithm.

| | | Noise | | | Blur | | | | Weather | | | | Digital | | | | | Acc. | Cache | |
| Alg. | BS | Gauss. | Shot. | Impul. | Defoc. | Glass. | Motion | Zoom. | Snow | Frost | Fog | Bright. | Contr. | Elast. | Pixel. | JPEG | Orig. | Avg | Avg | Max |
|---|---|---|---|---|---|---|---|---|---|---|---|---|---|---|---|---|---|---|---|---|
| | | | | | | | | CIFAR10-C | | | | | | | | | | | | |
| BN | 32 | 80.2 | 82.0 | 78.7 | 91.2 | 79.2 | 89.9 | 91.0 | 86.9 | 86.6 | 84.5 | 91.9 | 88.8 | 84.2 | 86.8 | 81.8 | 93.2 | 86.1 | 34 | 34 |
| BN | 64 | 81.3 | 82.7 | 79.6 | 91.9 | 79.9 | 90.8 | 91.6 | 87.5 | 87.0 | 85.5 | 92.3 | 89.4 | 85.3 | 87.5 | 82.6 | 93.7 | 86.8 | 67 | 67 |
| BN | 128 | 81.5 | 83.2 | 79.9 | 92.4 | 80.7 | 91.1 | 92.1 | 87.9 | 87.7 | 85.7 | 92.7 | 89.8 | 85.6 | 87.6 | 83.0 | 94.2 | 87.2 | 134 | 134 |
| TENT | 8 | 68.8 | 46.3 | 16.4 | 12.8 | 11.8 | 8.9 | 9.9 | 10.7 | 10.3 | 10.2 | 10.3 | 10.2 | 10.1 | 10.0 | 10.2 | 10.2 | 16.7 | 114 | 114 |
| | 16 | 81.8 | 82.2 | 71.8 | 68.9 | 52.8 | 43.7 | 39.0 | 30.7 | 22.8 | 17.7 | 10.1 | 6.7 | 7.4 | 8.2 | 8.1 | 9.0 | 35.1 | 229 | 229 |
| | 32 | 86.0 | 87.5 | 84.1 | 88.8 | 81.2 | 85.5 | 86.6 | 85.5 | 86.6 | 83.5 | 88.5 | 87.0 | 83.0 | 85.7 | 80.7 | 87.2 | 85.5 | 457 | 457 |
| +MECTA | 16 | 86.6 | 86.3 | 79.1 | 82.8 | 73.3 | 76.3 | 76.9 | 72.6 | 71.5 | 67.3 | 70.6 | 64.5 | 60.4 | 60.5 | 56.1 | 60.4 | 71.6 | 49 | 67 |
| +MECTA | 31 | 86.5 | 87.2 | 81.3 | 88.3 | 78.9 | 84.1 | 85.8 | 81.3 | 80.8 | 77.4 | 82.1 | 77.4 | 74.9 | 76.6 | 71.7 | 77.5 | 80.7 | 93 | 129 |
| EATA | 8 | 74.5 | 70.9 | 66.6 | 69.7 | 55.5 | 54.6 | 47.2 | 38.4 | 33.1 | 35.1 | 35.2 | 18.8 | 12.2 | 14.5 | 13.3 | 8.5 | 40.5 | 114 | 114 |
| | 16 | 83.4 | 84.3 | 81.3 | 86.4 | 77.0 | 83.8 | 86.5 | 84.5 | 84.7 | 83.9 | 88.1 | 87.9 | 80.9 | 85.5 | 79.3 | 87.9 | 84.1 | 229 | 229 |
| | 32 | 85.7 | 87.6 | 85.7 | 89.8 | 82.5 | 88.3 | 90.0 | 88.2 | 88.8 | 88.7 | 91.4 | 91.0 | 85.5 | 89.5 | 85.3 | 90.6 | 88.0 | 457 | 457 |
| +MECTA | 16 | 86.1 | 86.4 | 81.0 | 86.9 | 80.3 | 84.9 | 87.4 | 83.8 | 85.3 | 85.5 | 88.3 | 86.3 | 83.2 | 86.4 | 82.5 | 88.6 | 85.2 | 53 | 68 |
| +MECTA | 31 | 86.6 | 88.3 | 84.4 | 89.2 | 82.4 | 87.4 | 89.3 | 86.5 | 87.6 | 86.4 | 89.6 | 88.0 | 85.1 | 87.7 | 83.1 | 90.3 | 87.0 | 102 | 130 |
| | | | | | | | | CIFAR100-C | | | | | | | | | | | | |
| BN | 32 | 56.2 | 57.1 | 54.7 | 70.5 | 56.3 | 68.6 | 70.2 | 63.0 | 63.5 | 56.6 | 71.9 | 68.0 | 62.2 | 64.8 | 56.7 | 73.8 | 63.4 | 34 | 34 |
| BN | 64 | 56.8 | 58.7 | 55.8 | 71.5 | 57.7 | 69.5 | 71.5 | 64.1 | 64.3 | 57.4 | 73.1 | 69.0 | 63.6 | 66.1 | 58.4 | 75.3 | 64.6 | 67 | 67 |
| BN | 128 | 57.6 | 59.0 | 56.6 | 72.5 | 58.2 | 69.9 | 71.8 | 64.7 | 64.8 | 57.9 | 73.5 | 69.8 | 64.3 | 66.7 | 58.6 | 75.8 | 65.1 | 134 | 134 |
| TENT | 8 | 53.0 | 53.9 | 46.8 | 50.4 | 31.2 | 29.8 | 23.8 | 14.4 | 10.1 | 6.7 | 5.9 | 3.5 | 4.0 | 3.7 | 3.6 | 3.5 | 21.5 | 114 | 114 |
| | 16 | 56.8 | 61.4 | 59.6 | 68.5 | 56.1 | 65.3 | 66.9 | 59.7 | 60.0 | 54.9 | 65.0 | 57.0 | 54.2 | 56.2 | 46.8 | 58.6 | 59.2 | 229 | 229 |
| | 32 | 58.5 | 63.0 | 61.6 | 71.8 | 60.3 | 69.7 | 71.8 | 65.1 | 66.3 | 60.9 | 72.4 | 68.7 | 64.7 | 67.8 | 59.7 | 74.3 | 66.0 | 457 | 457 |
| +MECTA | 16 | 60.6 | 63.0 | 59.8 | 71.7 | 59.4 | 67.9 | 69.7 | 61.2 | 61.3 | 55.0 | 67.9 | 59.5 | 59.3 | 61.0 | 52.3 | 67.0 | 62.3 | 44 | 68 |
| +MECTA | 31 | 58.8 | 61.2 | 58.2 | 73.2 | 60.7 | 71.4 | 73.4 | 65.8 | 66.7 | 60.1 | 73.6 | 68.4 | 65.5 | 67.3 | 59.1 | 75.1 | 66.2 | 77 | 130 |
| EATA | 8 | 52.1 | 54.4 | 53.2 | 65.4 | 51.7 | 63.9 | 64.8 | 59.3 | 58.8 | 54.0 | 66.6 | 63.3 | 56.4 | 61.4 | 52.9 | 70.1 | 59.3 | 114 | 114 |
| | 16 | 57.3 | 60.5 | 58.5 | 69.9 | 57.1 | 68.9 | 69.8 | 63.7 | 64.4 | 59.4 | 71.6 | 67.9 | 62.8 | 67.1 | 58.2 | 74.4 | 64.5 | 229 | 229 |
| | 32 | 58.4 | 62.4 | 60.9 | 72.1 | 59.5 | 70.3 | 72.4 | 66.3 | 66.5 | 62.2 | 74.4 | 70.8 | 65.3 | 69.4 | 61.0 | 76.9 | 66.8 | 457 | 457 |
| +MECTA | 16 | 59.6 | 62.2 | 59.3 | 73.5 | 60.7 | 71.5 | 73.4 | 66.7 | 66.9 | 61.3 | 74.9 | 71.3 | 66.2 | 68.8 | 61.0 | 77.5 | 67.2 | 44 | 68 |
| +MECTA | 31 | 58.7 | 60.7 | 57.8 | 72.7 | 59.4 | 70.9 | 73.6 | 65.8 | 66.8 | 60.4 | 74.7 | 70.7 | 66.0 | 68.4 | 60.6 | 77.7 | 66.6 | 75 | 130 |
| | | | | | | | | ImageNet-C | | | | | | | | | | | | |
| BN | 64 | 38.4 | 41.6 | 38.8 | 29.1 | 32.1 | 40.1 | 46.4 | 44.1 | 46.8 | 54.8 | 67.7 | 35.1 | 53.7 | 64.7 | 54.6 | 73.5 | 47.6 | 206 | 206 |
| BN | 128 | 39.2 | 42.6 | 39.6 | 29.9 | 32.9 | 40.8 | 47.4 | 45.0 | 47.7 | 55.8 | 68.5 | 36.0 | 54.8 | 65.4 | 55.7 | 74.2 | 48.5 | 411 | 411 |
| TENT | 16 | 43.3 | 46.1 | 42.8 | 25.8 | 14.8 | 5.0 | 1.3 | 0.7 | 0.7 | 0.7 | 0.8 | 0.6 | 0.7 | 0.7 | 0.7 | 0.7 | 11.6 | 711 | 711 |
| | 32 | 46.1 | 50.6 | 49.2 | 37.2 | 38.2 | 41.2 | 42.5 | 36.6 | 36.2 | 38.8 | 46.8 | 27.5 | 30.5 | 34.3 | 26.9 | 34.8 | 38.6 | 2845 | 2845 |
| +MECTA | 16 | 48.6 | 49.3 | 46.6 | 33.6 | 34.6 | 34.8 | 37.5 | 28.9 | 26.7 | 29.2 | 35.7 | 14.0 | 20.6 | 21.9 | 15.1 | 19.3 | 31.0 | 194 | 212 |
| +MECTA | 30 | 48.6 | 50.9 | 48.5 | 35.7 | 38.3 | 39.6 | 44.2 | 37.0 | 37.4 | 42.1 | 51.9 | 31.7 | 42.9 | 47.6 | 42.5 | 53.6 | 43.3 | 338 | 397 |
| EATA | 16 | 44.4 | 47.1 | 45.4 | 39.0 | 39.4 | 47.4 | 49.7 | 49.7 | 48.4 | 57.6 | 64.3 | 47.8 | 54.5 | 61.7 | 56.3 | 69.5 | 51.4 | 711 | 711 |
| | 32 | 49.0 | 52.3 | 51.1 | 44.4 | 45.2 | 52.3 | 55.1 | 54.0 | 52.7 | 61.3 | 67.6 | 52.7 | 58.8 | 65.3 | 60.4 | 72.3 | 55.9 | 2845 | 2845 |
| +MECTA | 16 | 49.9 | 52.6 | 50.7 | 44.3 | 45.3 | 52.1 | 55.6 | 53.8 | 53.1 | 62.5 | 69.3 | 51.9 | 60.6 | 67.6 | 61.9 | 74.2 | 56.6 | 194 | 212 |
| +MECTA | 30 | 50.6 | 53.3 | 51.7 | 44.7 | 46.1 | 52.2 | 56.1 | 53.4 | 53.0 | 62.0 | 68.9 | 52.9 | 60.4 | 67.1 | 61.7 | 73.6 | 56.7 | 342 | 397 |

Table 7: Ablation study of MECTA-B on ImageNet-C with the highest severity level 5 regarding accuracy (%) and a batch size of 16. Blue cells highlight the accuracy that is the highest among all methods, and the bold texts indicate the best accuracy among ablations of EMA and MECTA-B.

| Alg. | EMA $\beta = 0.1$ | MECTA-B auto $\beta$ | Noise Gauss. | Shot. | Impul. | Blur Defoc. | Glass. | Motion | Zoom. | Weather Snow | Frost | Fog | Bright. | Digital Contr. | Elast. | Pixel. | JPEG | Orig. | Avg |
|---|---|---|---|---|---|---|---|---|---|---|---|---|---|---|---|---|---|---|---|
| BN | ✗ | ✗ | 33.7 | 36.5 | 34.0 | 24.6 | 27.2 | 35.0 | 40.5 | 39.1 | 42.3 | 49.4 | 62.7 | 30.7 | 47.6 | 59.1 | 48.9 | 69.1 | 42.5 |
| | ✓ | ✗ | 38.7 | 42.0 | 39.1 | 29.2 | 32.4 | 40.3 | 47.2 | 44.4 | 47.4 | 55.4 | 68.3 | 35.4 | 54.3 | 65.2 | 55.2 | 74.1 | 48.0 |
| | ✓ | ✓ | **39.8** | **43.3** | **40.3** | **30.3** | **33.4** | **41.1** | **48.3** | **45.2** | **48.3** | **56.6** | **69.1** | **36.0** | **55.4** | **66.1** | **56.2** | **74.7** | **49.0** |
| Tent | ✗ | ✗ | 43.3 | 46.1 | 42.8 | 25.8 | 14.8 | 5.0 | 1.3 | 0.7 | 0.7 | 0.7 | 0.8 | 0.6 | 0.7 | 0.7 | 0.7 | 0.7 | 11.6 |
| | ✓ | ✗ | 49.2 | 53.3 | 51.6 | 38.8 | 37.8 | 40.1 | 40.3 | 32.5 | 29.4 | 28.1 | 31.4 | 10.1 | 11.8 | 10.2 | 4.4 | 6.0 | 29.7 |
| | ✓ | ✓ | **50.3** | **54.6** | **53.0** | **40.7** | **40.2** | **42.3** | **42.9** | **35.3** | **33.0** | **32.2** | **36.7** | **14.3** | **15.9** | **15.2** | **8.5** | **11.8** | **32.9** |
| EATA | ✗ | ✗ | 44.4 | 47.1 | 45.4 | 39.0 | 39.4 | 47.4 | 49.7 | 49.7 | 48.4 | 57.6 | 64.3 | 47.8 | 54.5 | 61.7 | 56.3 | 69.5 | 51.4 |
| | ✓ | ✗ | 49.2 | 52.0 | 50.6 | 43.8 | 45.1 | 52.5 | 55.5 | 54.4 | 53.5 | 62.3 | 69.6 | 52.1 | 60.0 | 67.4 | 61.6 | 74.4 | 56.5 |
| | ✓ | ✓ | **50.0** | **53.1** | **51.5** | **44.8** | **45.7** | **53.2** | **56.5** | **55.3** | **54.3** | **63.2** | **70.3** | **52.6** | **61.1** | **68.1** | **62.5** | **75.0** | **57.3** |

Table 8: Evaluation of $k$-new $K$-old shift accuracy by EATA. Average accuracy (AA %) and worst accuracy (WA %) are reported for each target perturbation. Values in the brackets denote the difference between the current method and the base method using batch statistics.

| $K$ $k$ | EMA $\beta = 0.1$ | MECTA-B Auto $\beta$ | Impul. AA | WA | Motion AA | WA | Fog AA | WA | Elast. AA | WA |
|---|---|---|---|---|---|---|---|---|---|---|
| 49 1 | ✗ | ✗ | 35.5 | 34.6 | 37.0 | 36.8 | 50.1 | 49.8 | 48.3 | 47.8 |
| | ✓ | ✗ | 35.4 (**-0.1**) | 30.6 (-4.0) | 26.6 (-11.0) | 20.1 (-16.7) | 41.0 (-9.1) | 25.9 (-23.9) | 42.8 (**-5.5**) | 39.3 (-8.5) |
| | ✓ | ✓ | 34.4 (-1.1) | 32.0 (**-2.6**) | 28.6 (**-8.4**) | 25.7 (**-11.1**) | 43.6 (**-6.5**) | 39.1 (**-10.7**) | 42.6 (-5.7) | 41.2 (**-6.6**) |
| 9 1 | ✗ | ✗ | 34.1 | 33.4 | 35.7 | 35.6 | 49.4 | 48.9 | 47.4 | 47.1 |
| | ✓ | ✗ | 32.8 (-1.3) | 27.0 (-6.4) | 25.0 (-10.7) | 17.3 (-18.3) | 38.4 (-11.0) | 23.9 (-25.0) | 41.1 (-6.3) | 35.7 (-11.4) |
| | ✓ | ✓ | 34.1 (**0.0**) | 32.3 (**-1.1**) | 27.4 (**-8.3**) | 23.1 (**-12.5**) | 42.4 (**-7.0**) | 36.5 (**-12.4**) | 41.9 (**-5.5**) | 40.0 (**-7.1**) |
| 45 5 | ✗ | ✗ | 35.4 | 34.6 | 36.9 | 36.5 | 50.3 | 49.3 | 48.5 | 47.7 |
| | ✓ | ✗ | 38.4 (+3.0) | 37.5 (+2.9) | 35.4 (-1.5) | 30.6 (-5.9) | 53.3 (+3.0) | 50.2 (0.9) | 52.0 (+3.5) | 50.8 (+3.1) |
| | ✓ | ✓ | 37.8 (+2.4) | 37.2 (+2.6) | 35.4 (-1.5) | 33.0 (**-3.5**) | 51.5 (+1.2) | 49.6 (+0.3) | 50.8 (+2.3) | 49.6 (+1.9) |

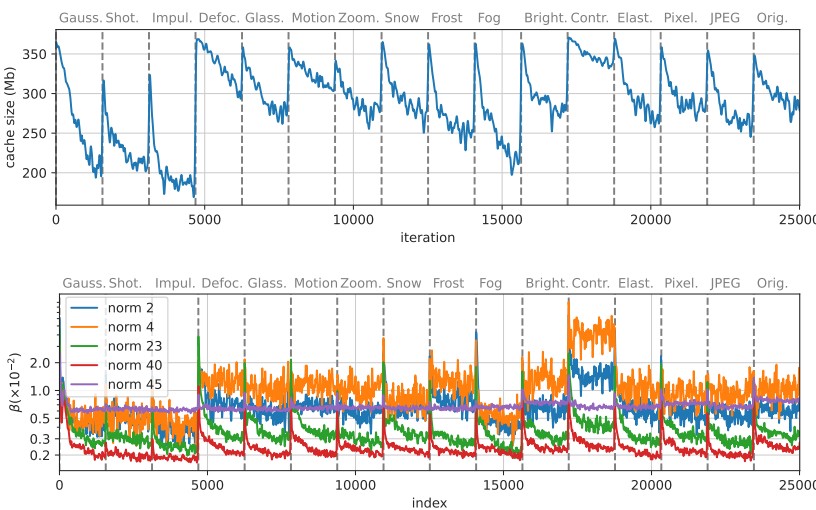

Figure 6: Dynamic cache size and $\beta$ using MECTA on ImageNet-C.

