# OpenReview forum: "MECTA: Memory-Economic Continual Test-Time Model Adaptation"
_ICLR.cc/2023/Conference — ICLR 2023 poster_

### Official Review · Reviewer_wXpt · 2022-10-20

**Confidence:** 4
**Correctness:** 4
**Technical Novelty And Significance:** 3
**Empirical Novelty And Significance:** 3
**Recommendation:** 8

**Clarity, Quality, Novelty And Reproducibility:**

Clarity: The paper is mostly well-written with a clear structure. The problem is well-motivated. The method is well explained.

Quality: The paper is of high quality, which can be reflected by good writing, solid problem motivation, simple but effective methodology formulation, and sound results.

Novelty: Both the problem and method are somewhat new.

Reproducibility: Given the pseudo codes in Alg. 1, and details on implementation and hyper-parameters, reproduction seems to be doable.

**Details Of Ethics Concerns:**

The methods do not use training data and do not store data during training. Thus, it does not cast risks to privacy or other aspects of ethics.

**Strength And Weaknesses:**

Strength:

1. The paper provides sufficient insights and motivations on continual test-time adaptation with memory efficiency.

2. The proposed methodology seems effective. On ImageNet dataset trained by ResNet50, it significantly reduces memory costs by at least 70% without sacrificing accuracy.

3. The proposed MECTA is efficient and can be seamlessly plugged into SOTA CTA algorithms at negligible overhead on computation and memory.

Weakness:

1. How this method can be extended to other normalizations? Could you elaborate more on this with examples and experiments?

2. More discussions on the potential applications of MECTA, in reality, are expected to better highlight its significance.

3. Table 1 is a bit hard to interpret, maybe highlighting the dataset names would be better.

4. The choices for the backbone network used in the experiments should be explained. Experiments on more network backbones (in particular large networks) would be interesting.


**Summary Of The Paper:**

The state-of-the-art adaptations improve out-of-distribution model accuracy via computation-efficient online test-time gradient descents but meanwhile cost about times of memory versus the inference, even if only a small portion of parameters are updated. To address this issue, the authors provide a novel solution called MECTA to drastically improve the memory efficiency of gradient-based CTA. MECTA can improve the memory efficiency of different CTA methods. The simple norm layer structure is ready to be plugged into various networks to replace batch-normalization. The MECTA Norm layer also enables the stop and restart of model adaptation without unused or absent caches for unwanted or on-demand back-propagation. To avoid forgetting due to removing parameters, pruning is conducted on cache data for back-propagation rather than forwarding and can greatly reduce memory consumption. Compelling results show that MECTA can maintain comparable performance to full back-propagation methods while significantly reducing the dynamic and maximal cache overheads.

**Summary Of The Review:**

More could be discussed about the implications of their results, but overall, this is a solid paper tackling an important problem. The proposed approach is interesting and the technique is sound. The claims are well supported by impressive empirical results.

---

> ### Author Response · Authors · 2022-11-18
> **Responses to Reviewer wXpt**
>
> We thank you for the positive feedback and below we tried our best to reply your questions.
>
> > Weakness:
> > **W1**: How this method can be extended to other normalizations? Could you elaborate more on this with examples and experiments?
>
> We appreciate the inspiring question on the normalization structure. Admittedly, our method is specialized for batch normalization, since recent advances in test-time adaptation have demonstrated the advantages of BN-based architectures in terms of accuracy and efficiency.
>
> Here, we exemplify the extensibility of the channel pruning of MECTA by layer normalization. The major difference between batch and layer normalization is that the batch dimension is decoupled from the normalization. The gradient for the affine parameters are
> $$\sum_{n=1}^B \frac{\partial \ell_n}{\partial \gamma_i^l} = \sum_{n=1}^B \sum_{j=1}^W\sum_{k=1}^H\frac{\partial \ell_n}{\partial a^l_{i,j,k}} z^l_{n,i,j,k}, $$
> following the notations of Eq. (3). The n, i, j, k are the indexes of the batch, channel, height and width dimensions. Therefore, we can drop the channel dimensions of $z^l$ stochastically to save memory.
>
>
> > **W2**: More discussions on the potential applications of MECTA, in reality, are expected to better highlight its significance.
>
> MECTA can be used for on-device model adaptation, especially for large models. In Appendix A.1, we provided several examples where the on-device memory is limited, e.g., on Raspberry Pi and mobile phones. These devices can only afford inference tasks. Therefore, MECTA can be used to adapt large models on device for better test-time performance. Except for mobile devices, MECTA can be applied for self-driving to robustify models in dynamic environments, like snow and rainy weather.
>
> Beyond image classifications, the principles of MECTA (especially stochastic cache) can be applied to parameter-efficient finetuning of large language models based on low-rank adaptation (LoRA). In LoRA, the update to a weight matrix is decomposed into low-rank matrixes. Computing the corresponding gradients also requires the cache of the input features. Following the idea of MECTA, we first sparsify the vector gradients and then drop the corresponding cache values.
>
> > **W3**: Table 1 is a bit hard to interpret, maybe highlighting the dataset names would be better.
>
> We have updated Table 1 with grey backgrounds to highlight the datasets.
>
> > **W4**: The choices for the backbone network used in the experiments should be explained. Experiments on more network backbones (in particular large networks) would be interesting.
>
> In Table 1, our choice is based on the ‘robustbench’ benchmark (https://robustbench.github.io/), which ranks methods on CIFAR10, CIFAR100, and ImageNet-C by robust accuracy against common corruptions. In Table 2, we follow the prior baselines, EATA and Tent, to use the ResNet50 for a fair comparison.
>
> Thanks for the valuable suggestion of additional experiments. In Table 5, we extend our experiments to other network structures, from small networks (MobileNet to huge models, like VGG19 and ResNeXt. In most architectures, our method not only requires lower cache sizes but also achieves higher accuracy.

---

> > ### Comment · Reviewer_wXpt · 2022-11-20
> > **Response to authors**
> >
> > Thanks for addressing my comments and questions. I keep my original rating.

---

> > > ### Author Response · Authors · 2022-11-29
> > > **Thank you!**
> > >
> > > Thanks for your responses and reviews.

---

### Official Review · Reviewer_pnAa · 2022-10-22

**Confidence:** 4
**Correctness:** 4
**Technical Novelty And Significance:** 4
**Empirical Novelty And Significance:** 3
**Recommendation:** 8

**Clarity, Quality, Novelty And Reproducibility:**

Clarity: Most parts of the paper are clearly written and explained. Minor concerns on clarity:
* The forget gate is not clearly stated. The authors may mention the term in Alg. 1.
* In Table 1, EATA is not always the best method on all datasets. On CIFAR100, Tent outperforms EATA. An explanation should be provided.
* In Page 9, the sentence before (B.1) does not fit into the content. The authors may explain this.

Quality: The studied problem is interesting and practical, none of the prior works investigated it before. This paper is technically solid. The algorithm design is delicate. Both efficiency and accuracy during continual test-time adaptation are well backed by compelling results.

Novelty: The initiated study on the memory efficiency of continual test-time adaptation is a novel problem. The proposed methodology also seems novel.

Reproducibility: The algorithm pseudo codes are enclosed in the main body. Implementation and hyper-parameters details are provided in Appendix B.1. The datasets and baselines codes are all public available online and properly specified in the paper.


**Details Of Ethics Concerns:**

The methods do not use training data and do not store data during training. Thus, it does not cast risks to privacy or other aspects of ethics.

**Strength And Weaknesses:**

Strength:

1. The authors initiate a pilot study on the memory efficiency and reveal the main bottleneck is on the intermediate results cached for back-propagation, even if only few parameters need to be updated in the state-of-the-art computation-efficient solution (EATA).
2. The proposed MECTA is very well motivated. MECTA can drastically improve the memory efficiency of gradient-based CTA.
3. The organization and writing are very clear. The authors did a good job in reviewing all the related work, and summarizing the corresponding advantages and disadvantages.
4. The proposed MECTA is backed by very impressive empirical results. Given limited memory constraints, MECTA improves the Tent and EATA by 8.5 − 73% accuracy on CIFAR10-C, CIFAR100-C, and ImageNet-C datasets.

Weakness:

1. In addition to the current related work section, I highly recommend a discussion on model adaptation from big pre-trained models on device, and explain why it’s not a viable direction.
2. Can the authors report the computation complexity in terms of FLOPs?
3. I understand that all the three dimensions (Reduce B, Reduce C, Dynamic L) play quite important roles in MECTA, I also want to know which dimension is the dominant one and why.


**Summary Of The Paper:**

This paper proposes MECTA in order to improve out-of-distribution model accuracy via computation-efficient online test-time gradient descents in a memory economic manner. The key idea behind MECTA is to reduce batch sizes, adopt an adaptive normalization layer to maintain stable and accurate predictions, and stop the back-propagation caching heuristically. The networks are first pruned to reduce the computation and memory overheads in optimization and the parameters are then recovered afterward to avoid forgetting. MECTA can be seamlessly plugged into state-of-the-art CTA algorithms at negligible overhead on computation and memory. On three datasets, CIFAR10, CIFAR100, and ImageNet, MECTA improves the accuracy by at least 8.5% with constrained memory and significantly reduces the memory cots of ResNet50 on ImageNet by at least 70% without sacrificing accuracy.

**Summary Of The Review:**

Overall, this paper is technically strong and novel. It solves an important problem in continual test-time adaptation with solid methodology and sound experiments.

---

> ### Author Response · Authors · 2022-11-18
> **Responses to Reviewer pnAa**
>
> We thank you for the positive feedback, and please find our responses below.
>
> > Weakness:
> > **W1** : In addition to the current related work section, I highly recommend a discussion on model adaptation from big pre-trained models on device, and explain why it’s not a viable direction.
>
> Thank you for the suggestion. We believe model adaptation from big pre-trained models for changing test-time environments is possible. In Table 5, we extend our experiments to other network structures, from small networks (MobileNet) to huge models, like VGG19, ResNeXt, and ResNet152. Our method outperforms EATA with much lower cache sizes and higher accuracy. Especially, in Fig.4, we studied the effect of network depth in detail, where we scale the network up to 152 layers. We show that memory improvement increases by depth.
>
> > **W2**: Can the authors report the computation complexity in terms of FLOPs?
>
> We updated Table 1 with FLOPs for each method. Our method only marginally increases FLOPs by less than 0.2%.
>
> > **W3**: I understand that all the three dimensions (Reduce B, Reduce C, Dynamic L) play quite important roles in MECTA, I also want to know which dimension is the dominant one and why.
>
> We conducted qualitative studies in Section 5.2 to fully understand the roles of the three dimensions. In Fig.2, we observe that the channel pruning most significantly pushes the accuracy-memory frontier forward. The stochastic pruning of caches or gradients can maintain a good performance with a large pruning ratio, 70% here, because the continual adaptation imputes the missing gradients sequentially.
>
> > Minor concerns on clarity:
>
> > * In Table 1, EATA is not always the best method on all datasets. On CIFAR100, Tent outperforms EATA. An explanation should be provided.
>
> Though Tent outperforms EATA, the difference is marginal. As demonstrated in Table 2, EATA is specialized in terms of computation efficiency, since much fewer samples are used for back-propagation.

---

### Official Review · Reviewer_veMt · 2022-10-23

**Confidence:** 5
**Correctness:** 4
**Technical Novelty And Significance:** 4
**Empirical Novelty And Significance:** 3
**Recommendation:** 8

**Clarity, Quality, Novelty And Reproducibility:**

Clarity: The work provides a clear motivation by visualizing the huge memory consumption in Figure 1 which is hard to adapt to small edge devices. Then the method development is clearly based on the bottleneck revealed by Eq (3) in terms of the batch size, the channel number, and the network depth.

Quality: The major claims including efficiency and adaptation accuracy have been well demonstrated by experiments. Specifically, the authors compare the model performance under the same cache constraint where MECTA significantly improves the accuracy of baselines. The benchmarks also show that MECTA mitigates catastrophic forgetting in Tent even without sample selection (like EATA). Without memory constraint, MECTA can greatly reduce memory consumption yielding comparable accuracy as the state-of-the-art baseline.

And the effects of each component in the algorithms are well explored. For example, layer-wise training complies with the non-uniform forgetting preference by layers (Fig. 3), and adaptive memorization can improve the per-domain accuracy and mitigate accuracy drops on shift. The new-designed shift accuracy precisely reveals how the accumulated memorization (in BN) affects the accuracy of domain shift. The ablation study in terms of accuracy-memory trade-off also helps understand how each component improves the frontiers.

Novelty: The paper considers a rather novel challenge: reduce memory consumption at test-time adaptation. The proposed method introduces a parameter-free memorization mechanism, layer-wise training strategy, and random cache pruning, which are novel in the scope of continual test-time adaptation. The authors also defend the technical novelty by comparing the technique to traditional memory footprint reduction, at the end of Page 5.

Reproducibility: The method is enclosed into one layer, as stated in Alg. 1. The major steps include computing distribution divergence, moving average, randomly dropping cache channels, and cache on demand. These steps are elementary and could be easily implemented. But I still appreciate it if the authors can publish codes (upon acceptance).


**Details Of Ethics Concerns:**

The methods do not use training data and do not store data during training. Thus, it does not cast risks to privacy or other aspects of ethics.

**Strength And Weaknesses:**

Strength:

1. The investigated problem is significant to real IoT scenarios. The authors reveal the overlooked low memory efficiency of traditional CTA methods, which is critical for applications on resource-constrained edge devices. Especially when considering the Internet of Things, the memory-limited tiny edge devices cannot afford 4Gb+ memory consumption for test-time adaptation as demonstrated by Fig 1.

2. The technical part is novel. The proposed method is motivated by Eq. (3) that the parameter-efficient training (only updating BN layers) still brings in large memory caches for back-propagation. Based on the motivation, the authors provide intuitive solutions that can effectively adapt models without large memory consumption in three dimensions.

    - For reducing batch sizes, the proposed method can stabilize the training using an adaptive memorization rate (beta) which is parameter-free and intuitive for the dynamic statistical shifts. The intuition is well supported by prior observations that the BN statistics differ by domain.

    - For reducing channels in the cache, the random dropping and on-demand training can avoid forgetting resembling implicit regularization when the momentum of SGD or Adam can make up the missing gradients.

    - The layer-wise training is intuitive as the adaptation typically is on demand of domain shift.

3. The intuition of these methods are well supported by experiments. For instance, the use of BN-based distribution divergence as the indicator is supported by the experiment in Fig 3. The forgetting mitigation of MECTA is supported by the benchmarks on Tent.

4. I appreciate the authors designed the shift-accuracy evaluation which can reveal the significant impact of accumulating batches in the transition of domains. Except for the plausible results of MECTA, the experiment itself could remind other researchers to carefully handle the domain shift at test time instead of focusing on in-domain average accuracy.

Weakness:

1. The authors should discuss the connection between the proposed adaptation methods and existing parameter-efficient fine-tuning works. For example, the low-rank adaptation of transformers [A].

2. The authors only consider the ResNet and should discuss how the method can be extended to other model architectures.

[A] LoRA: Low-Rank Adaptation of Large Language Models. Edward J. Hu*, Yelong Shen*, Phillip Wallis, Zeyuan Allen-Zhu, Yuanzhi Li, Shean Wang, Lu Wang, Weizhu Chen


**Summary Of The Paper:**

Continual test-time model adaptation (CTA) recently draws researchers' attention since trained models may encounter dynamically-changing test-time environments. This work first revealed the limitations of prior arts in memory efficiency which could be a critical obstacle for applications of CTA to memory-limited edge devices and proposed a novel algorithm that can significantly reduce the memory overhead at test time. The proposed algorithm enables test-time adaptation using smaller batches, and sparse layer-and-channel caches without significantly losing accuracy. The authors empirically demonstrated the outstanding accuracy and memory efficiency of the proposed method.

**Summary Of The Review:**

The paper revealed and addressed an overlooked yet an important problem in continual test-time adaptation. Most claims are well supported by experiments and the methods are intuitive and well-explained. The experiments are detailed and clear. Thus, I recommend acceptance.

---

> ### Author Response · Authors · 2022-11-18
> **Responses to Reviewer veMt**
>
> We thank you for the positive feedback, and below are our responses to your questions.
>
> > Weakness:
> >
> > **W1**: The authors should discuss the connection between the proposed adaptation methods and existing parameter-efficient fine-tuning works. For example, the low-rank adaptation of transformers [A].
>
> In the robustbench (https://robustbench.github.io/), which benchmark models using common corruptions, the ResNet models are still state-of-the-art architectures on robust accuracy. Therefore, we focus on the ResNet backbone in our work. We believe that it is an interesting direction to use transformers in our future work.
>
> > **W2**: The authors only consider the ResNet and should discuss how the method can be extended to other model architectures.
>
> In Table 5, we extend our experiments to other network structures, from small networks (MobileNet to huge models, like VGG19, ResNeXt. Since our method only modifies the batch-normalization layers, it can be easily plugged into these networks. Our method outperforms EATA with much smaller cache sizes and higher accuracy.

---

### Official Review · Reviewer_QbXv · 2022-10-24

**Confidence:** 4
**Correctness:** 3
**Technical Novelty And Significance:** 2
**Empirical Novelty And Significance:** 3
**Recommendation:** 6

**Clarity, Quality, Novelty And Reproducibility:**

Clarity:

- The exposition clearly identifies the dimensions controlling memory usage—(B)atch size, (C)hannels, and (L)ayers—and adequately explains how each is reduced by the proposed method. The core technical content of the paper is clear enough.
- Proofreading is needed throughout including in the abstract. The meaning can be determined, but the writing errors do make reading more effortful.
- The main contributions of this work could be better highlighted and named. MECTA norm is not a new norm, but follows the use of EMA by Yang et al. 2022 (cited) and Schneider et al. 2020 (uncited in the method section, though cited in related work). The point of calling it a normalization layer seems more to indicate the desired interface to the method, which is to incorporate each part into a substitute layer for batch normalization.

Quality:

- The claims of memory usage are grounded and measured by considering peak memory usage and its different components during inference and adaptation (Sec. 1, cache measurements in the main results of Tables 1 & 2).
- The chosen benchmarks and experimental design are standard and established by prior work such as EATA, so the results can be understood, and they can be compared with existing papers without additional effort.
- The analysis experiments (Section 5.2) check key properties such as the use of the adaptive statistics threshold auto $\beta$, and the results support the design choices of MECTA.


Novelty:

- This work is not the first to emphasize computational efficiency during test-time adaptation. Tent made a first step by only requiring one forward and backward per point, and EATA further considered when and when not to update. Nevertheless, this work provides the first detailed empirical study of memory usage in the context of test-time adaptation.
- The masking of gradients during backward is related to forward masking like Dropout and DropConnect, but nevertheless it is different. - To the best of my knowledge this is a new use of stochastic masking during optimization. It is a simple trick, but it does have a use in controlling memory usage, and it serves its purpose in the experiments.

Reproducibility:

- This work should be reproducible. It makes use of common datasets in an already-defined experimental setting, the method section (Sec. 4) is sufficiently detailed, and the code will be released.


Miscellaneous Feedback

- For Table 4, the "K, k" notation for old and new batches is hard to parse. Consider alternative letters like O for number of Old batches and N for number of New batches.


**Strength And Weaknesses:**

Strengths

- MECTA does reduce the peak memory usage of Tent and EATA while still improving accuracy on shifted data in the continual adaptation setting. MECTA allows for 4x larger batches and consequently improves the accuracy of EATA by more than 10 points. The accuracy of Tent is also improved, but Tent was not designed for continual adaptation in the same way as EATA, so this improvement is less surprising and significant.
- The stochastic dropping of channel-wise gradients ("Sparse gradients via stochastically-pruned caches") is a novel scheme to reduce gradient memory as far as I am aware. Dropping gradients instead of pruning channels has the advantage of preserving the forward computation and accuracy. (However, see weaknesses for a discussion of whether or not this can truly save memory given how deep networks are implemented.)
- The layer-wise gating of updates is more granular than the input-wise gating of EATA, and so more computation can potentially be spared. In particular, MECTA can stop caching as soon as a layer is gated, while EATA requires an entropy prediction and therefore a full forward (and the corresponding memory) for each input to decide its gating.
- MECTA has a regularizing side-effect on Tent that helps prevent forgetting, as shown by improved accuracy on _unshifted_ data after adaptation. This follows from its sparser updates and moving estimates.
- The implementation of MECTA is encapsulated in a variation on the batch normalization layer, which makes it easy to adopt by swapping it into a network with batch normalization.
- The experiments and in particular the analysis experiments in Figure 2 show that MECTA strictly dominates adaptation by BN for accuracy and memory usage. (However, without memory limits Tent and EATA can still do as well or better than MECTA, so this gain comes at the cost of specialization to this reduced computation setting.)

Weaknesses

- This work lacks simple computational baselines that would help prove the necessity and impact of MECTA's contributions. In particular, it does not investigate (1) updating fewer layers or (2) checkpointing gradients.
  - Fewer layers: this work ignores the option of simply updating fewer layers during testing, by for instance only updating the last/deepest layers of the network. Only updating deeper layers removes the need for caching of all earlier layers.
  - Gradient checkpointing: this work ignores the option of recomputation (a.k.a. rematerialization) by simply discarding forward caches and recomputing forward as needed for backward, and by doing so misses a simple baseline. This is now a common feature of deep learning frameworks, as is provided by the gradient checkpointing utility in PyTorch, for example.
- A prior paper on continual adaptation is missing from the related work: CoTTA at CVPR'22. CoTTA is a contemporary method from the same time as EATA, and so likewise deserves inclusion in the related work on continual adaptation. In the submission, it is only mentioned and disqualified for its computational cost in the experiments (Section 5), which is not appropriate.
- The proposed channel-sparse gradients may not save time or memory in practice. Most frameworks only support dense gradients for inputs and parameters, so zeroing out a particular channel may not alter the computation performed.
The analysis experiments or "Qualitative Studies" of Section 5.2 are done with a subset of the common corruptions. These fidings may or may not generalize to the full set, so it would be better to do them with the full set, for thoroughness and comparability with other results.


**Summary Of The Paper:**

Test-time model adaptation updates model parameters during inference in order to reduce generalization error on shifted data. Continual test-time adaptation, the setting of this work, does so for varying shifts without knowledge of when the shift itself changes over time. The purpose of this work is to improve the efficiency of inference and adaptation to require less computation time and memory. In particular, the proposed memory economical continual test adaptation (MECTA) approach extends and tunes entropy minimization methods (Tent and EATA) so that their gradient updates do not need as much memory as the naive implementation of backpropagation during testing. This work highlights the memory usage of caching forward activations in particular, which is measured to be 5x the memory for inference with at ImageNet scale (ResNet-50, specifically). Efficiency is improved while maintaining accuracy by reducing batch size, extending normalization statistics to moving estimates rather than batch-wise estimates, dropping channels by a kind of test-time pruning or partial caching that is sparser than the standard backward pass, and gating updates by thresholding the change in layer-wise statistics. Any layer that is gated does not need its cache for updates, and so memory can be saved for the gated layer and the following layers. MECTA improves accuracy over Tent and EATA when memory is constrained (Table 1) while nearly achieving the same accuracy as EATA when memory is not limited (Table 2). These results are shown for the standard choices of datasets: ImageNet-C and CIFAR-10/100-C with common baseline models like ResNet-50.

**Summary Of The Review:**

The deployment of test-time adaptation does require efficiency so that the updates made during inference do not delay predictions too much. Even more seriously, the adaptation computation needs to be feasible at all by fitting in device constraints such as memory, which is the constraint addressed by this work. Precise measurements of memory are provided, and the proposed MECTA offers several techniques that combine to reduce peak memory usage, at least in principle. The reduction of batch size and the gating of layer updates should reduce memory in practice also. On the other hand, the pruning/masking of gradient channels may or may not reduce memory depending on the implementation, so this requires clarification. While the experiments show the effect of the proposed techniques, they do not cover simple baselines, such as updating fewer layers or applying gradient checkpointing.

The amount of memory saved can be as much as ~70% of the original usage for a ResNet-50, but the lack of computational baselines makes it unclear how much this amount matters.

Questions:

- How much memory is saved and accuracy maintained by simply updating different numbers of layers in the network, starting with the deepest and then adding more shallow layers? Please Consider only a single layer, say 10% of the layers, and 50% of the layers.
- What is the memory/time trade-off of test-time adaptation with gradient checkpointing when compared to MECTA? In particular, what is this trade-off when limited by the memory constraints applied to Table 1?
- Does the channel sparsity of the gradient truly achieve reduced computation in practice, that is on a GPU with CUDA, or is there only a potential reduction?
- Does the dynamic cache truly reduce peak memory usage, or only average memory usage? The analysis results of Figure 3 suggest that the peak usage remains high, at the beginning of each shift. Please confirm the effect of MECTA on the peak usage.

**Update following Response & Discussion**: The response addressed the questions and weaknesses identified by this review adequately, but not totally, and on the balance this deserves a recommendation that sides with acceptance. I have raised the score to borderline accept accordingly. The results from the response for simpler computational baselines justify the need for MECTA, and the further details provided by the response and revision make it more informative. MECTA is worth considering for acceptance because of its more dynamic and memory-efficient computation for adaptation.

---

> ### Author Response · Authors · 2022-11-18
> **[1/2] Responses to Reviewer QbXv**
>
> We appreciate your detailed review and constructive suggestions. We tried our best to reply as follows, where we denote the points mentioned in weakness as W1, W2, etc, and questions as Q1, Q2, etc.
>
> Weaknesses
> > **W1&Q1,Q2**: Missing baselines: (1) updating fewer layers or (2) checkpointing gradients, (3) CoTTA.
>
> We appreciate the suggested baselines and have added them to our paper. Below is a detailed discussion comparing the baselines with our method.
>
> (1) In Fig. 5 of revision in Appendix B.2, we evaluate EATA with fewer trainable layers, which reduces the cache size. During adaptation, we keep $k$ deepest layers to be trained and freeze other layers, which is denoted as L$k$ in the figure. We also include the gradient checkpointing (EATA+GC) for the trainable layers. We see that reducing the trainable layers significantly decreases the cache size, which is even lower than MECTA. However, the corresponding accuracy is significantly decreased by 5% compared to EATA+MECTA meanwhile. In comparison, though MECTA also uses layer-sparse training, our method presents the best accuracy-memory trade-off in the experiment. The key difference is that our method sparsifies the training only on demand, specifically when a layer is well adapted without the need for further training.
>
> (2) We add gradient checkpointing (GC) as a baseline to Table 1 and 2, where we present the GFLOPs as the metric of hardware-independent time complexity. In Table 1, though GC compresses the cache of MECTA without affecting accuracy, it meanwhile increases the computation costs by around $20\%$ in terms of GFLOPs. In contrast, our method only marginally increases the computation cost by less than $0.2\%$ with better accuracy and can bear larger batches with even smaller caches. In Table 2, we also compare the memory reduction given the same batch size where the GC cache size is in the brackets. Notice that GC is less effective in cache reduction here than the reported results by (Chen, et al., 2016) because the parameter-efficient adaption already drops a lot of caches for frozen convolutional layers. In comparison, MECTA only costs 50% cache of GC and 30% of EATA with comparable performance.
>
> (3) We add CoTTA to Table 2. Though CoTTA outperforms Tent, it requires 32 more times of forward computations. EATA+MECTA has a much lower cache size and computation overheads (in terms of the number of forwards).
>
> We hope our response has adequately addressed your concerns.
>
> > **W2&Q3**: The proposed channel-sparse gradients may not save time or memory in practice on a GPU with CUDA.
>
> We believe it is important that MECTA reduces memory costs in practice, and we argue that our method can reduce memory and maintain time efficiency at the same time.
> * For the **time** complexity, in Table 1, we show the number of float-point operations where our method only increases the FLOPs by 0.1% compared to the baseline.
> * For the memory, in practice, we do not need to cache the zeroed caches (not gradients) for back-propagation since the zeroed gradients do not need a cache. Therefore, we can save time and computation for the sparse gradients simultaneously. We add the **details of implementations at the end of Appendix B.1**. During forwarding, we only store the remained values of $z^l$ and the indexes of the remained channels (denoted as $R$). Later, we compute the gradient on the **remaining channels only** and apply the updates to corresponding channels defined by $R$. The implementation effectively reduces memory consumption with a small extra space for storing the index set $R$.
>
> In addition, we conduct experiments to show that on GPU, our method reduces memory more than gradient checkpointing. On an NVIDIA A5000 GPU, our method reduces the running memory from 7.6Gb to 5.5Gb for ResNet50 with a batch size of 64. Note that for NVIDIA GPU, there is a large amount of memory reserved for efficiency considerations of inference, which can be optimized during the deployment.
>
> > **W3**: The analysis experiments or "Qualitative Studies" of Section 5.2 are done with a subset of the common corruptions. These findings may or may not generalize to the full set.
>
> To solidify the experiments, we follow your valuable suggestion to extend the experiment in Fig 2 to the full set of corruptions, where the new results do not change our conclusions, though. Note that Fig 3 and Table 3 had already been conducted on the full set in our initial submission.

---

> > ### Comment · Reviewer_QbXv · 2022-11-28
> > **Thank you for the response to the weaknesses and questions.**
> >
> > As the response has adequately addressed the negative points raised in the initial review, I am raising score to side with acceptance.
> >
> > > W1&Q1,Q2: Missing baselines: (1) updating fewer layers or (2) checkpointing gradients, (3) CoTTA.
> >
> > The response adequately addresses all points. I suggest including the computational baselines (training fewer layers, gradient checkpointing) in the paper either in the main text or in the appendices with a mention in the main text. Checking that these baselines are not sufficient on their own strengthens the case for MECTA.
> >
> > > W2&Q3: The proposed channel-sparse gradients may not save time or memory in practice on a GPU with CUDA.
> >
> > The response mostly addresses these points, but not entirely. In particular I would like to emphasize _time_ and not FLOPs, as the memory operations for channel sparsity are not measured by FLOP counts. That is, on a GPU, the indexing and updating of channels (as well as the allocation of an intermediate, dense array of channels for gradient computation) may take more time than the non-sparse computation. However, this point about time is not critical since FLOPs and memory are measured, but I would advise the authors to also report time for completeness (perhaps in an appendix, if need be).
> >
> > > Q4: Does the dynamic cache truly reduce peak memory usage, or only average memory usage? The analysis results of Figure 3 suggest that the peak usage remains high, at the beginning of each shift. Please confirm the effect of MECTA on the peak usage.
> >
> > The response addresses this point, and Fig. 3 confirms reduced memory usage for the peak and average.
> >
> > > Questions on novelty: MECTA norm is not a new norm, but follows the use of EMA by Yang et al. 2022 (cited) and Schneider et al. 2020 (uncited in the method section, though cited in related work).
> >
> > The response addresses this point, and the clarification about the dynamic beta is helpful. It may be worthwhile to highlight this dynamic (re-)estimation of beta in the introduction or method sections.

---

> > > ### Author Response · Authors · 2022-11-29
> > > **Thank you for your updates**
> > >
> > > We appreciate your positive comments and very constructive feedback. We will update our final revision based on your comments.

---

> ### Author Response · Authors · 2022-11-18
> **[2/2] Responses to Reviewer QbXv**
>
> > **Q4**: Does the dynamic cache truly reduce peak memory usage, or only average memory usage? The analysis results of Figure 3 suggest that the peak usage remains high, at the beginning of each shift. Please confirm the effect of MECTA on the peak usage.
>
> Yes. Our method reduces memory usage, both on the peak and on average. For clarification, we updated Fig. 3, where we depict two baselines, EATA and EATA+GC (gradient checkpointing). Our peak usage is lower than the best baseline, EATA+GC, when we can even dynamically reduce the memory costs.
>
> > Questions on novelty: MECTA norm is not a new norm, but follows the use of EMA by Yang et al. 2022 (cited) and Schneider et al. 2020 (uncited in the method section, though cited in related work).
>
> We have updated our paper with the missing citation (before Eq. 4), and we argue here that our method is distinguished from the previous works. One essential difference between our norm and the previous EMA is that the forgetting rate (or momentum coefficient, beta) is adaptively determined by the current statistics per layer. The adaptivity is essentially motivated by the dynamics of the statistics by time and by layer. We demonstrate such dynamics in Fig 3 (the lower figure), where the estimated beta varies by time (due to varying environments) and layer (due to asynchronous adaptation paces). A constant beta (Yang, et al., 2022) cannot accommodate such dynamics. A monotonically decreasing beta (by the number of samples) (Schneider et al., 2020) cannot make it, either, since a new environment requires an increased beta rather than a decreased one. Therefore, we propose a novel dynamic beta estimation. The adaptive beta also enables the early stopping of layers, which was never explored before.

---

> ### Author Response · Authors · 2022-11-24
> **A gentle reminder**
>
> Dear Reviewer QbXv,
>
> We want to express our appreciation for your detailed comments and insights again. This is a gentle reminder that we have updated the paper and tried our best to address your concerns.
>
> In the responses, we have provided a point-to-point response to your concerns. Please kindly let us know if you have any concerns you find not fully addressed. We would be more than happy to include your suggestions in the update.
>
> Best,
>
> Authors

---

### Decision · Program_Chairs · 2023-01-20

**Decision:**

Accept: poster

**Justification For Why Not Higher Score:**

The reason a rating of Accept (poster) is justified. The paper is addressing an important problem and the reviewers have a consensus about its acceptance. However, as also pointed out in the reviews, the paper isn't the first to look at the problem of memory-efficient test-time adaptation; prior works such as TENT and EATA have also looked at this issue. Therefore, as considering the various other points made by the reviewers in their reviews and discussion, it was decided to not give a higher score than Accept (poster).

**Justification For Why Not Lower Score:**

The paper definitely is above the bar for acceptance.

**Metareview: Summary, Strengths And Weaknesses:**

This paper studies the problem of lifelong/continual test-time adaptation and specifically looks at methods that perform an online gradient descent at test-time. Such methods tend to be memory-intensive. The paper proposes several strategies to yield significant memory savings.

In the initial reviews, this paper received highly positive and detailed reviews from 3 of the 4 reviewers. The 4th reviewer also gave a detailed review but expressed some concerns about the lack of comparisons with simple baselines. During the rebuttal period, the authors conducted and reported these experiments as well, after which the 4th reviewer also raised the score to 6.

There is a clear consensus among the reviewer for acceptance. The paper addresses an important problem at the intersection of lifelong/continual learning and test-time adaptation, proposes novel techniques, and has extensive experimental evaluation. Therefore, I recommend the paper for acceptance.

**Note From Pc:**

if the above contains the word "oral" or "spotlight" please see: "oral" presentation means -> notable-top-5% and "spotlight" means -> notable-top-25%. As stated in our emails, we are disassociating presentation type from AC recommendations